# Modeling early pathophysiological phenotypes of diabetic retinopathy in a human inner blood-retinal barrier-on-a-chip

Thomas L. Maurissen [1], Alena J. Spielmann [1], Gabriella Schellenberg[1], Marc Bickle[2], Jose Ricardo Vieira[1], Si Ying Lai[1], Georgios Pavlou[3], Sascha Fauser[1], Peter D. Westenskow [1], Roger D. Kamm [3,4] & Héloïse Ragelle [1]

Diabetic retinopathy (DR) is a microvascular disorder characterized by inner blood-retinal barrier (iBRB) breakdown and irreversible vision loss. While the symptoms of DR are known, disease mechanisms including basement membrane thickening, pericyte dropout and capillary damage remain poorly understood and interventions to repair diseased iBRB microvascular networks have not been developed. In addition, current approaches using animal models and in vitro systems lack translatability and predictivity to finding new target pathways. Here, we develop a diabetic iBRB-on-a-chip that produces pathophysiological phenotypes and disease pathways in vitro that are representative of clinical diagnoses. We show that diabetic stimulation of the iBRB-on-a-chip mirrors DR features, including pericyte loss, vascular regression, ghost vessels, and production of pro-inflammatory factors. We also report transcriptomic data from diabetic iBRB microvascular networks that may reveal drug targets, and examine pericyte-endothelial cell stabilizing strategies. In summary, our model recapitulates key features of disease, and may inform future therapies for DR.

Preclinical research is progressively shifting towards human-based models using tissue-specific primary cells or stem cell derivatives[1,2]. Bioengineered human model systems closely mimic the physiological organization and function of native tissues, enabling researchers to investigate disease mechanisms and response to treatments[3]. Several approaches have been developed to model 3D vasculature[4–11], including tissue-specific microvasculatures of the retina[12], brain[13,14] and placenta[15]. Current models are commonly used to assay acute changes in permeability or target expression in response to a disease trigger or treatment, and there are no models exploring chronic effects on vascular morphology and pathway dysregulation in long-term cultures. Such models are critical to elucidate the pathogenesis of

microvascular disorders that affect proper tissue function of most organs in different disease contexts.

Diabetic retinopathy (DR) is one of the main chronic microvascular complication of diabetes, affecting over 100 M patients worldwide[16–18], and causes regressive alterations in capillaries of the inner retina. The inner blood-retinal barrier (iBRB) is a highly selective endothelial barrier that maintains tissue homeostasis by regulating permeability and molecular transport between the blood circulation and neural retina. Disruption of the iBRB in DR patients leads to progressive retinal neurodegeneration and loss of vision[19].

In early stages of the disease (non-proliferative diabetic retinopathy, NPDR), a series of microvascular injuries including basement

[1]Roche Pharma Research and Early Development, Cardiovascular, Metabolism, Immunology, Infectious Diseases and Ophthalmology, Roche Innovation Center Basel, F. Hoffmann-La Roche Ltd, Basel, Switzerland. [2]Roche Pharma Research and Early Development, Institute of Human Biology, Roche Innovation Center Basel, F. Hoffmann-La Roche Ltd, Basel, Switzerland. [3]Department of Biological Engineering, Massachusetts Institute of Technology, Cambridge, MA, USA. [4]Department of Mechanical Engineering, Massachusetts Institute of Technology, Cambridge, MA, USA. ✉e-mail: rdkamm@mit.edu; heloise.ragelle@roche.com

membrane thickening, pericyte dropout, low grade inflammation and disruption of tight junctions causes regression of retinal capillaries, hypoxia, excessive vascular permeability and neovascularization (proliferative diabetic retinopathy, PDR)[20–22]. The precise sequence of NPDR pathological events is unknown. Furthermore, diabetic macular edema (DME) is a major feature of DR where fluid leakage causes retinal swelling[23]. Current therapies, such as intravitreal injections of anti-vascular endothelial growth factor (VEGF) or dual anti-VEGF/angiopoietin (ANG) 2 (Faricimab) promote retinal drying in DME patients and mitigate the progression and severity of PDR[24,25]. Additional therapeutic options targeting NPDR are needed. A more complete understanding of the pathophysiological mechanisms underlying NPDR is critical to develop novel therapies and relevant human model systems that are capable of modeling these changes in vitro are lacking[26].

The in vitro study of iBRB function, disease and injury requires the physiological integration of relevant retinal microvascular cell types into functional tissue units that express characteristic iBRB markers and respond to disease triggers[26,27]. In the retinal microvasculature, pericytes and astrocyte endfeet envelop capillaries and regulate endothelial barrier properties. These interactions become disrupted in DR pathophysiology. Several angiogenic communication pathways exist between pericytes and endothelial cells (ECs) that present therapeutic potential including, but not limited to, the ANG/TIE2 pathway[28]. Several organ-on-a-chip technologies have been leveraged for ophthalmology applications to model the choroid, the outer BRB, the interface between photoreceptors and RPE, or investigate pericyte-EC interactions[29–33]. There is currently no human model system to study pathophysiological changes in cell types of the iBRB. The models of the iBRB that have been developed so far are based on flat Transwell inserts where ECs, pericytes and/or astrocytes are seeded on a polycarbonate membrane or at the bottom of the well[34–36]. While these Transwell iBRB models have highlighted the contribution of pericytes and astrocytes to endothelial barrier properties, they lack an extracellular matrix, direct physical contact between cell types and a vascular-like architecture. Reconstructing these specialized features of the iBRB in vitro and interrogating them experimentally will enable better understanding of iBRB properties in physiological and disease conditions as well as provide a platform for validating therapeutic approaches.

In this work, we developed an iBRB-on-a-chip model by assembling human primary retinal microvascular cells into perfusable microvascular networks (MVNs). The iBRB-on-a-chip mimicked native tissue organization and relevant expression of iBRB markers. Experimental diabetic stimulation triggered NPDR-associated changes, including loss of pericytes, vascular regression and the formation of ghost vessels, which correspond to clinical hallmarks in patients with NPDR[23,37]. Morphological changes were quantified using high content confocal imaging and automated image analyses and transcriptomic profiles were characterized over time. Differential expression levels confirmed the implication of pro-inflammatory and vascular instability pathways in the diabetic iBRB-on-a-chip, and targeting pericyte-EC interactions induced treatment-specific phenotypes. Data from this model may inform pathogenesis and innovative therapeutic strategies.

## Results

### Inclusion of perivascular cell types improves structural properties of the iBRB-on-a-chip

We made the model retina-specific and clinically relevant by co-culturing primary human retinal microvascular ECs (HRMVECs) with primary human retinal microvascular pericytes (HRPs) and primary human retinal astrocytes (HRAs) at 1:1:1 ratio in a fibrin gel, since each of these three cell types undergoes pathophysiological changes in retinal microvascular diseases. We selected a 1:1 EC to pericyte ratio to be consistent with the ratio found in human retinas[38] and we added a similar astrocyte fraction based on the literature[13,39–41]. Cells were cultured in medium supplemented with 50 ng ml⁻¹ VEGF for 4 days to initiate iBRB MVN formation. The VEGF concentration was then reduced to a basal level (5 ng ml⁻¹) in order to stimulate quiescence and induce barrier properties between day 4 and 7 (Fig. 1a and Supplementary Fig. 1a). Confocal imaging showed self-organization of retinal cells into agglutinin positive (UEA I+) endothelial networks with co-localized PDGFRβ+ pericytes and S100b+ astrocytes (Fig. 1b, c and Supplementary Fig. 1b), similar to the cellular interactions and physiological architecture of the iBRB. Cross-sections revealed the formation of perfusable lumina (Fig. 1d and Supplementary Fig. 1c). Compared to tri-culture conditions, mono-culture of ECs led to fewer CD31+ network interconnections, possibly due to early vascular regression, and co-culture with pericytes resulted in a dense network of larger CD31+ endothelial tubes (Fig. 1e, f and Supplementary Fig. 1d, e). Addition of astrocytes reduced the vascular area and diameter, making the networks more microvascular-like (Supplementary Fig. 1f, g). These changes suggested a functional role of astrocytes in regulating microvascular morphology[42,43], which supported using a tri-culture system. Replacing HRMVECs with primary human umbilical vein ECs (HUVECs) in tri-cultures resulted in similar microvascular morphology. In sum, both perivascular cell types, pericytes and astrocytes, positively affected EC organization as well as MVN dimensions and connectivity. Importantly, when HRMVECs from different donors were co-cultured with HRPs and HRAs, all HRMVEC donors formed 3D networks (Supplementary Fig. 2). These results highlight that HRMVECs in our tri-culture system formed 3D MVNs independent of the donor.

### Critical barrier features are recapitulated in the iBRB-on-a-chip

We confirmed the maturity and functionality of iBRB MVNs by validating the expression of tight junction (CLDN5 and ZO-1) and adherens junction (VE-cadherin) proteins that are characteristic of a tight iBRB (Fig. 2a). We observed close physical interactions between ECs and pericytes (Supplementary Fig. 3). Similarly, ECM proteins (LAM and COL IV), components of the endothelial basement membrane, were produced around UEA I+ tubes (Fig. 2b and Supplementary Fig. 4a). Inside ECM networks, we identified laminin+, and collagen type IV+ tubes that were partially UEA I- and thus non-vascularized. These ghost vessels started appearing after vascular remodeling in the MVN formation phase, and indicated vascular reorganization or early regression. Differential gene expression analysis further confirmed the expression of EC- (CD31 encoded by *PECAM1*, VE-cadherin encoded by *CDH5*, *VWF*, *CLDN5*, *OCLN*, ZO-1 encoded by *TJP1*), pericyte- (*PDGFRB*, NG2 encoded by *CSPG4*, *ACTA2*, *RGS5*, N-cadherin encoded by *CDH2*) and astrocyte-specific (*GFAP*, *S100B*, *SOX9*, *AQP4*) markers, as well as angiogenesis (*TIE1*, TIE2 encoded by *TEK*, *ANGPT1*, *ANGPT2*) and ECM (*COL1A1*, *COL4A1*, *LAMA1*) markers (Fig. 2c and Supplementary Fig. 4b). Gene expression levels varied during MVN formation and maturation phases, in response to VEGF supplementation. Tight junction claudin-5 (*CLDN5*) expression increased, suggesting enhanced barrier properties[13,44]. *ANGPT2*, a vascular destabilizing agent, and *COL4A1* expression peaked when supplementing VEGF, while *ANGPT1*, a quiescence inducing factor, and *LAMA1* decreased (Fig. 2c and Supplementary Fig. 4c, d), suggesting endothelial activation[45]. Next, to verify iBRB function, we perfused the MVNs with fluorescent-labelled dextran (70 kDa TRITC-dextran) and quantified vascular permeability (Fig. 2d–f). We obtained an apparent permeability value of approximately $1 \times 10^{-7}$ cm s⁻¹, which is comparable to values measured in vivo[46]. Importantly, permeability increased 4.7-fold when the barrier disruptor TNF-α was applied to the iBRB MVNs. Collectively, these data suggest that perfusable MVNs reliably represent a functional iBRB.

### Inducing pathophysiological phenotypes associated with diabetic retinopathy

To model microvascular changes caused by diabetes, we exposed iBRB MVNs to an elevated concentration of D-glucose (30 mM or

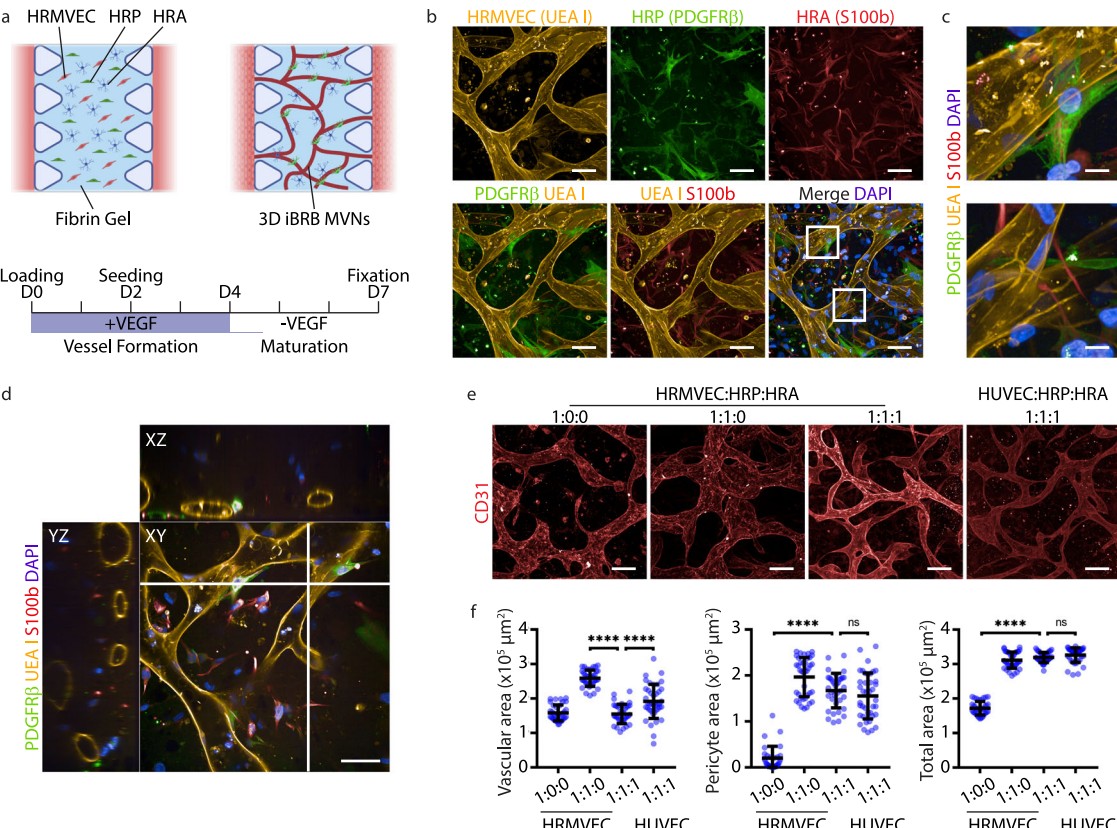

**Fig. 1 | Inner blood-retinal barrier model formed of self-assembled micro-vascular networks. a** Schematic (top) of 3D inner blood-retinal barrier (iBRB) microvascular network (MVN) formation: the MVNs are formed by mixing human retinal microvascular endothelial cells (HRMVECs, 6 × 10⁴ cells), human retinal pericytes (HRPs, 6 × 10⁴ cells) and human retinal astrocytes (HRAs, 6 × 10⁴ cells) in 10 µl of fibrin gel (3 mg ml⁻¹). Created with BioRender.com. Timeline (bottom) of cells in fibrin loaded in the devices and cultured for 4 days in medium supplemented with 50 ng ml⁻¹ VEGF (vessel formation) and 3 days in medium with basal VEGF (vessel maturation) to allow iBRB MVN formation. On D2, the side channels are seeded with HRMVECs (3 × 10⁴ cells per channel). **b** Representative images of iBRB MVNs: HRPs (PDGFRβ) and HRAs (S100b) co-localized near HRMVEC networks (UEA I). DAPI was used to visualize nuclei. Frames in the merged image indicate enlarged regions used in (**c**). **c** Enlarged image regions showing pericyte-EC interactions (top) and astrocyte protrusions in proximity to the abluminal endothelial surface (bottom). The images were taken on D7 and show maximum intensity projections of 124.5 µm Z-stacks. Stainings were repeated in *n* = 5 independent experiments with similar results. **d** Cross-section of iBRB MVNs on D7. Image shows orthogonal projections of 124.5 µm Z-stacks. Stainings were repeated in *n* = 5 independent experiments. **e** Representative images of endothelial networks (CD31) containing different HRMVEC:HRP:HRA cell number ratios. Images show maximum intensity projections of 290 µm Z-stacks. Experiments were repeated in *n* = 3 independent experiments. **f** Quantification of CD31+ vascular area (left), PDGFRβ+ pericyte area (middle) and F-actin+ total area (right). *n* = 35 HRMVEC 1:0:0, *n* = 39 1:1:0, *n* = 36 1:1:1 and *n* = 40 HUVEC 1:1:1 networks analyzed from *n* = 3 independent experiments. Data are mean ± s.d. ****P < 0.0001; one-way ANOVA. Source data are provided as a Source Data file. Scale bars, 10 µm (**c**), 50 µm (**b**, **d**) and 100 µm (**d**).

540 mg dl⁻¹; grade 4 serum hyperglycemia corresponds to >500 mg dl⁻¹) and pro-inflammatory cytokines TNF-α and IL-6 (1 ng ml⁻¹ each)[47]. Vitreous and plasma levels of TNF-α and IL-6 are elevated in patients with DR and these pro-inflammatory cytokines were added to the high glucose medium to mimic the inflammatory state observed in diabetes[48–51]. Diabetic iBRB MVNs were examined on day 7 (D7), 14 (D14) and 28 (D28) to identify NPDR phenotypic changes compared to controls (Fig. 3a) and image analyses were performed on maximum intensity projections and 3D reconstructions of image stacks of full-length iBRB MVNs (Supplementary Figs. 5–7). Quantified readouts included vascular area, pericyte coverage, ECM area, and ghost vessel fractions. The total vascular area of control iBRB MVNs decreased to approximately 70% between D7 and D14 before stabilizing until D28. On the other hand, diabetic iBRB MVNs regressed continuously to 30% of the initial control value until D28 (Fig. 3b, c and Supplementary Fig. 5a, b). Vascular regression with diabetic treatment was confirmed by quantifying EC nuclei (Supplementary Fig. 5c, d). Similarly, the pericyte area decreased to approximately 50% of the initial control value at the end of the diabetic treatment (Fig. 3d–f and Supplementary Fig. 6a). In order to verify if the loss of pericytes occurred irrespective of pericyte location or preferentially in

proximity of the vasculature, we quantified overlapping pericyte and EC areas. Using this analysis, we observed a 3-fold reduction in pericyte coverage between D7 and D28 in diabetic iBRB MVNs (Fig. 3g and Supplementary Fig. 6b–e). Finally, we investigated alterations of basement membrane deposition as another hallmark of NPDR. We observed that the collagen type IV+ area remained constant over time and across conditions (Fig. 3h–j). The fraction of ghost vessels, defined as the ratio of avascular collagen IV+ area to total collagen IV+ area, significantly increased with diabetic treatment (Fig. 3k and Supplementary Fig. 6f, g). The ghost vessel fraction increased 1.6-fold with diabetic treatment between D7 and D28, and increased 1.3-fold between control and diabetic conditions on D28. The major changes induced by the diabetic treatment appeared between D14 and D28, indicating the importance of maintaining chronic diabetic treatments in order to model corresponding phenotypic changes. By applying diabetic treatment onto the iBRB-on-a-chip, we modeled phenotypic changes corresponding to clinical hallmarks of NPDR.

Additionally, we investigated further effects of the diabetic treatment between D7 and D28. First, evaluation of the effects of each individual component of the diabetic cocktail compared to combined effects showed significant decrease of vascular area and pericyte

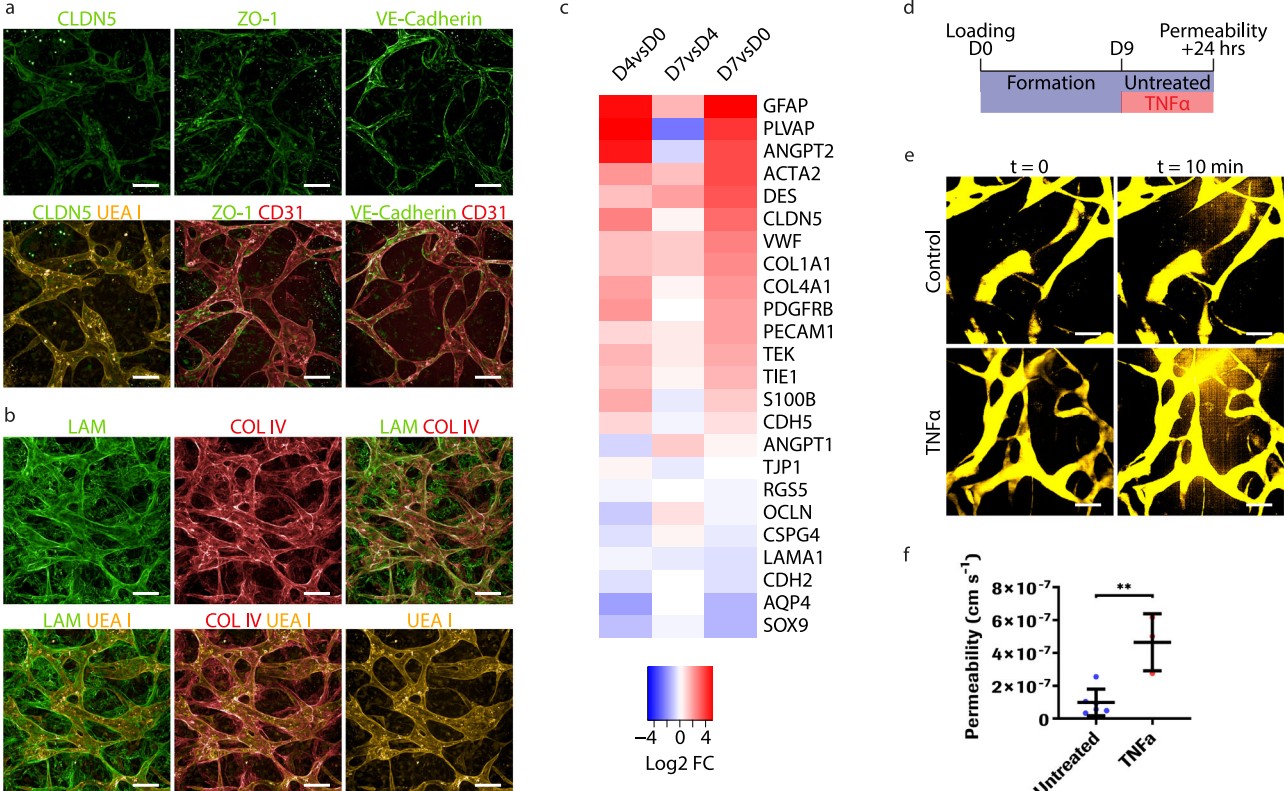

**Fig. 2 | Inner retinal microvasculature indicate mature and functional barrier properties. a** Representative images of tight junction (CLDN5 and ZO-1) and adherens junction (VE-cadherin) proteins co-localizing with endothelial networks (UEA I or CD31). Stainings were repeated in *n* = 3 independent experiments with similar results. **b** Basement membrane proteins (LAM and COL IV) co-localizing with endothelial networks (UEA I). Images show maximum intensity projections of 290 μm Z-stacks. Stainings were repeated in *n* = 3 independent experiments. **c** Heat map of differential gene expression between D4 and D0 (vessel formation), D7 and D4 (vessel maturation), and the whole iBRB MVN formation process between D7 and D0. Results are from a defined characterization gene panel without cutoff. Data are RNA-Seq aggregated Log2 FC from *n* = 3 independent

experiments. **d** Permeability assay timeline: iBRB MVNs were formed between D0 and D7, were either kept in standard culture or treated with 1 ng ml⁻¹ TNF-α for 24 h before performing the permeability assay. **e** Representative image of TNF-α-treated (1 ng ml⁻¹, 24 h) iBRB MVNs perfused on D10 with TRITC-labelled dextran (100 μg ml⁻¹, 70 kDa), acquired by fluorescence confocal microscopy at 5 min intervals. At t = 10 min, the binary mask shows leakage. Perfusion images are maximum intensity projections of 290 μm Z-stacks. **f** Apparent permeability coefficients were quantified on D10 in untreated (blue) and TNF-α-treated MVNs (red) using 70 kDa TRITC-dextran as a tracer. *n* = 6 untreated and *n* = 3 TNF-α-treated networks. Data are mean ± s.d. **P = 0.0029; two-tailed Student's t-test. Source data are provided as a Source Data file. Scale bars, 100 μm (**a**, **b**, **e**).

coverage, and increase of ghost vessels in diabetic conditions compared to glucose, IL-6, and TNF-α alone on D28 (Supplementary Fig. 8). These results demonstrate that the effects we observed in diabetic conditions are attributed to the combination of factors and not to the effect of one single component. Then, an apoptosis (Caspase-3/7) detection assay showed that viability remained above 95% in untreated and IL-6-treated conditions, and above 85% in diabetic conditions, indicating increased cell death (Supplementary Fig. 9). In order to provide insight into the sequence of events during initiation of NPDR, we co-stained apoptotic nuclei with endothelial UEA I and pericyte PDGFRβ markers. Apoptotic cells on D9, D11 and D14 during early treatment overlapped mostly with UEA I+ vascular networks, suggesting that ECs are the initial cell type undergoing apoptosis on D9, followed by pericytes from D11 (Supplementary Fig. 10a). The number of apoptotic ECs increased in all conditions, and was more than 2-fold greater in diabetic conditions (Supplementary Fig. 10b). Finally, we perfused untreated and diabetic MVNs with fluorescent-labelled dextran (70 kDa TRITC-dextran) on D14 and D28. While the current tri-culture setup was not perfusable at these later time points, we identified other astrocyte batches that supported long-term perfusion and produced similar diabetic phenotypes on D28 (Supplementary Fig. 11a, b). Diabetic treatment strongly reduced perfusability, the percentage of TRITC+ perfusable vessels compared to UEA I+ total vessels for all astrocyte batches (Supplementary Fig. 11c). In perfusable networks, we

quantified vascular permeability and did not observe increased leakage in perfusable diabetic vessels (Supplementary Fig. 11d). In sum, diabetic treatment is responsible for disease phenotypes, and causes increased cell death and decreased perfusability that contribute to iBRB dysfunction.

## Critical genes associated with the iBRB are altered in diabetic conditions

We performed RNA-sequencing (RNA-seq) to determine which transcriptomic changes were underlying the NPDR-associated phenotypic alterations identified in Fig. 3. Principal component analysis showed distinct clusters for diabetic-treated and control samples at different time points (Fig. 4a and Supplementary Fig. 12). Untreated controls and osmotic controls treated with D-Mannitol clustered together and showed no differential gene expression, indicating similarity. Diabetic treatment increased the number of down-regulated genes at the later time point (824 on D14 and 1346 on D28), while the number of up-regulated genes remained constant (456 on D14 and 462 on D28) (Fig. 4b, c). To extend the analysis, we plotted the data in heat maps according to the categories of vascular regression, pericyte loss, ECM changes, inflammation, and astrocytes (Fig. 4d–h). Vascular stability markers *ANGPT1* (−3.2 Log2 FC), *SERPINF1* (−1.2 Log2 FC) encoding PEDF and *CLDN5* (−1.0 Log2 FC) were downregulated in diabetic conditions (Fig. 4d). Pericyte markers such as *RGS5* (−2.3 Log2 FC), *ACTA2*

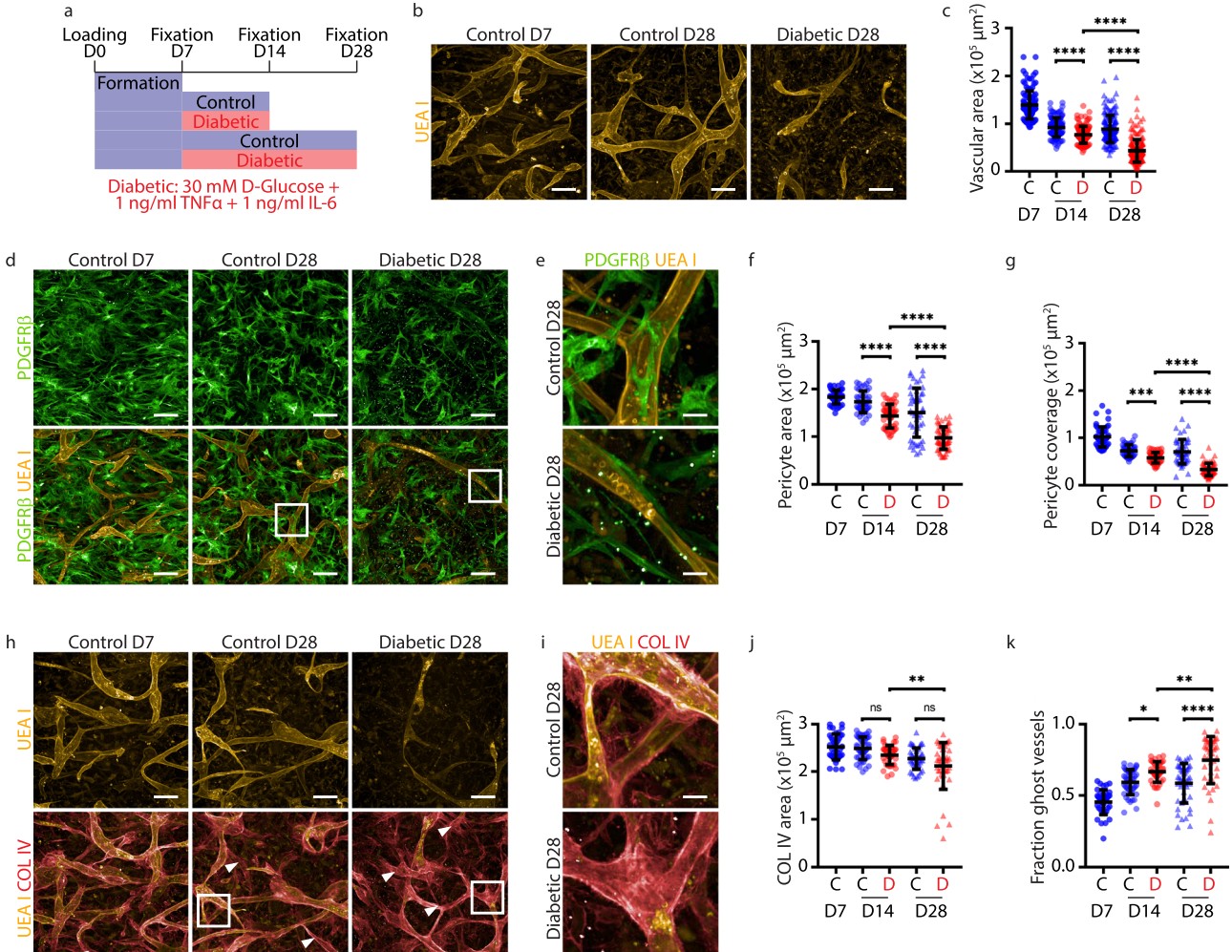

**Fig. 3 | Modeling diabetic retinopathy with chronic diabetic stimulation.**
**a** Timeline of disease modeling. **b** Representative images of iBRB MVNs (UEA I) cultured in standard medium on D7 (left), D28 (middle) or in diabetic medium on D28 (right). **c** Quantification of vascular area for control on D7, D14 (blue circles) and D28 (blue triangles) and diabetic conditions on D14 (red circles) and D28 (red triangles). $n = 113$ control D7, $n = 117$ D14, $n = 165$ D28, $n = 117$ diabetic D14 and $n = 203$ D28 networks analyzed from $n = 3$ independent experiments.
**d** Representative images of pericytes (PDGFRβ) and endothelial networks (UEA I). Frames in the merged images indicate enlarged regions used in (**e**). **e** Enlarged image regions showing pericyte-EC interactions. **f** Quantification of pericyte area. **g** Quantification of pericyte coverage defined by the pericyte area overlapping the vascular area. $n = 49$ control D7, $n = 42$ D14, $n = 48$ D28, $n = 48$ diabetic D14 and $n = 61$ D28 networks analyzed for pericyte area and coverage, from $n = 3$

independent experiments. ***$P = 0.0007$. **h** Representative images of endothelial networks (UEA I) and overlay images with basement membranes (COL IV) revealing the presence of ghost vessels that are COL IV+UEA I− (arrowheads). Frames indicate enlarged regions used in (**i**). **i** Enlarged image regions showing increased ghost vessels in diabetic conditions (bottom) on D28. **j** Quantification of collagen type IV area. **$P = 0.0045$. **k** Quantification of the fraction of ghost vessels calculated by the difference of COL IV+ area and vascular area, divided by the COL IV+ area. $n = 38$ control D7, $n = 45$ D14, $n = 45$ D28, $n = 40$ diabetic D14 and $n = 42$ D28 networks analyzed from $n = 3$ independent experiments. Data are mean ± s.d. *$P = 0.0270$; **$P = 0.0094$; ****$P < 0.0001$; one-way ANOVA. Source data are provided as a Source Data file. All images show maximum intensity projections of 395 µm Z-stacks. Scale bars, 20 µm (**e**, **i**) and 100 µm (**b**, **d**, **h**).

(−1.4 Log2 FC) and *PDGFRβ* (−1.3 Log2 FC) were downregulated in diabetic conditions while *PDK4* (2.6 Log2 FC) was upregulated (Fig. 4e), collectively suggesting pericyte dysfunction[52–54]. *CD248* (−4.4 Log2 FC) coding for endosialin was one of the most downregulated pericyte markers, supporting vascular instability[55]. Matrix metalloproteinases (MMPs) were upregulated[56], particularly *MMP3* (6.8 Log2 FC) and *MMP9* (2.0 Log2 FC) on D14 and *MMP9* (5.2 Log2 FC) and *MMP12* (4.5 Log2 FC) on D28 (Fig. 4f). *COL4A1* (1.0 Log2 FC) and *COL4A2* (1.1 Log2 FC) expression also increased, while *LAMA1* (−0.5 Log2 FC) and *LAMA2* (−1.5 Log2 FC) expression and other collagen types decreased. Furthermore, the expression levels of inflammation markers peaked on D14 including *IL1A* (3.5 Log2 FC), *CXCL8* (2.3 Log2 FC), *CCL2* (1.9 Log2 FC), *CCL7* (1.5 Log2 FC), *CXCL1* (1.5 Log2 FC), and *IL6* (1.0 Log2 FC), while *IL1B* (5.7 Log2 FC) expression further increased until D28 (Fig. 4g). Finally, similar to pericyte markers, astrocyte markers were

downregulated (Fig. 4h). In sum, these transcriptomic changes and corresponding pathways perturbations reflected the multifactorial effects caused by chronic diabetic treatment (Supplementary Figs. 13 and 14).

At the protein level, we detected increased levels of COL IV and MMP-9 in diabetic supernatants, in accordance with RNA-seq data (Fig. 4i, j and Supplementary Fig. 15). ANG1 decreased while ANG2 and receptor TIE2 did not change significantly (Fig. 4k–m and Supplementary Fig. 15). However, ANG2 levels were approximately 150-fold higher than ANG1, implying barrier instability. In addition, the inflammation markers CXCL1, ICAM-1, IL-1β, IL-6 and MCP-1 increased in supernatants of diabetic conditions, mostly at the earlier time point D14 (Fig. 4n–r and Supplementary Fig. 15), suggesting a sustained inflammatory response with chronic diabetic treatment consistent with RNA-seq data. Overall, these results mirror NPDR phenotypes.

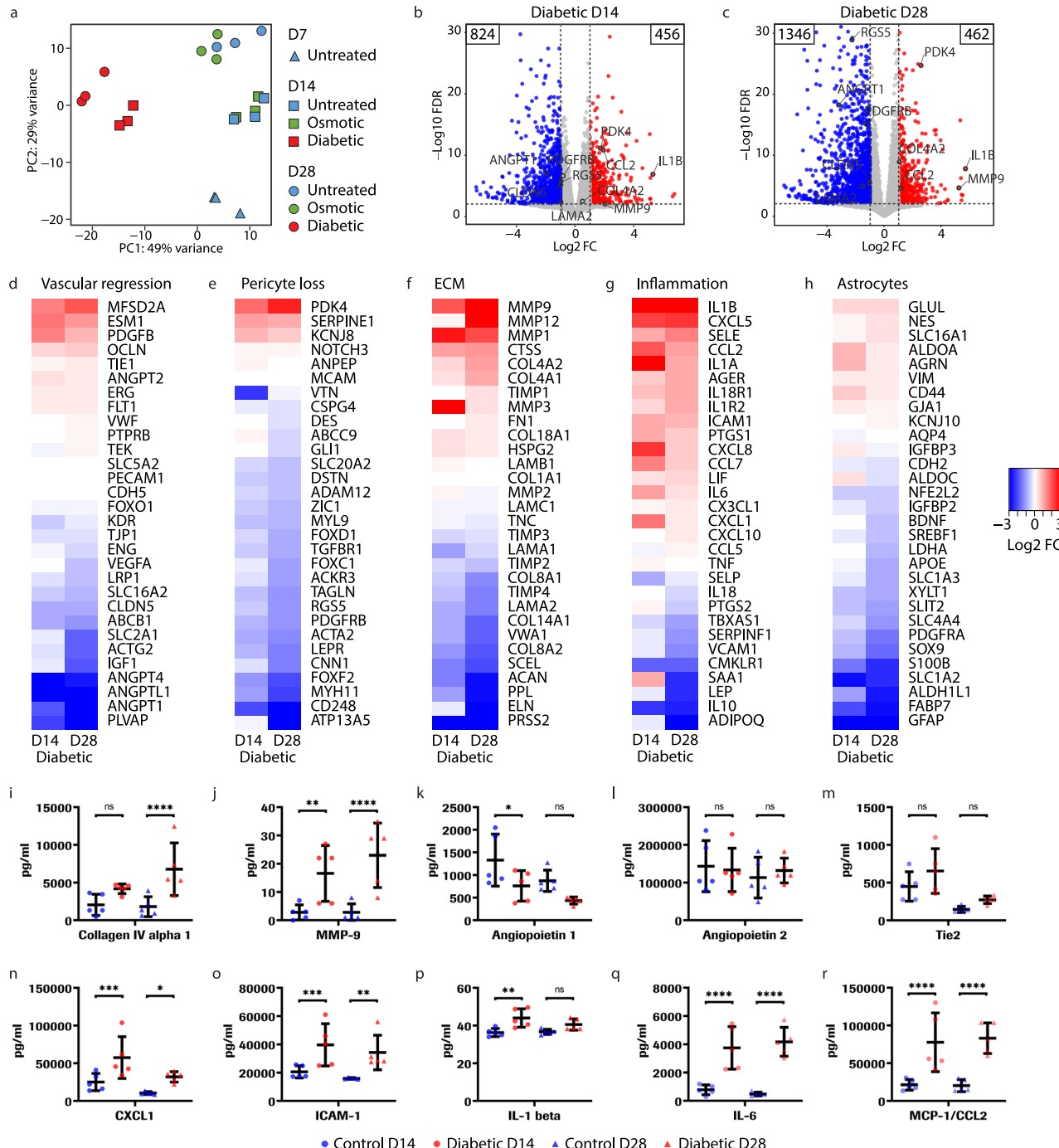

**Fig. 4 | Transcriptomic analysis reveals pathways affected in diabetic retinopathy. a** Principal component analysis of RNA-Seq data from untreated and osmotic controls and diabetic treatment. Data are from $n = 3$ independent experiments with each replicate shown. Volcano plots of differentially expressed genes (FC > 2, FDR < 0.01) of diabetic D14 (**b**) and D28 (**c**) compared to untreated controls at the same time points, with the number of up- and down-regulated genes indicated (FC > 2). Heat maps of differential gene expression between diabetic treated and untreated conditions corresponding to vascular regression (**d**), pericyte loss (**e**), ECM remodeling (**f**), inflammation (**g**) and astrocytes (**h**). Results are from defined gene panels without cutoff. Data are aggregated log2 FC from $n = 3$ independent experiments. Analyte measurements from untreated (blue) and diabetic treated (red) supernatants collected on D14 (circles) and D28 (triangles), showing concentrations of human collagen IV alpha I (**i**), MMP-9, **P = 0.0023 (**j**), Angiopoietin-1, *P = 0.0119 (**k**), Angiopoietin-2 (**l**), Tie-2 (**m**), CXCL1, *P = 0.0288, ***P = 0.0009 (**n**), ICAM-1, **P = 0.0014, ***P = 0.0009 (**o**), IL-1β, **P = 0.0011 (**p**), IL-6 (**q**) and MCP−1 (**r**). Supernatants are obtained from $n = 5$ independent experiments. Data are mean ± s.d. ****$P < 0.0001$; one-way ANOVA. Source data are provided as a Source Data file.

## Pericyte-endothelial cell interactions can be experimentally perturbed by targeting key signaling pathways

We next set out to target pericyte-EC communication pathways[57] specifically by inhibiting PDGFRβ in pericytes using an anti-PDGFRβ antibody (APB5), TIE2 in ECs using a TIE2 small compound inhibitor (Tie2i), and Notch receptor cleavage using a γ-secretase inhibitor (DAPT). We also applied 19,20-dihydroxydocosapentaenoic acid (DHDP) and advanced glycated end product (AGE) treatments expected to interfere with homeostatic cellular processes (Fig. 5a). DHDP can perturb pericyte-EC interactions and endothelial cell junctions by integrating in the cell membrane, inhibiting the catalytic unit of gamma-secretase and disrupting the localization of cadherins[58]. AGE

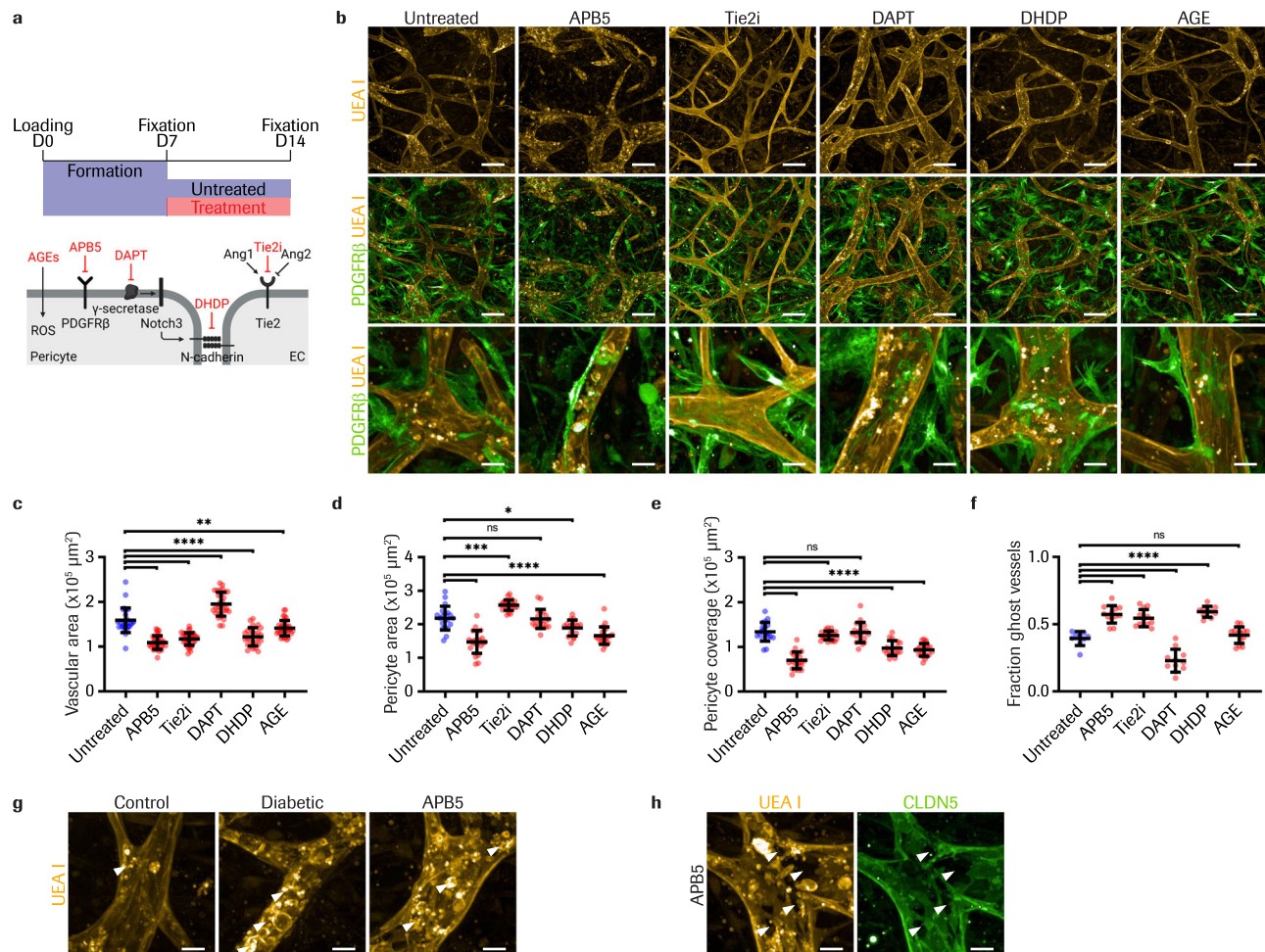

**Fig. 5 | Inhibition of pericyte-endothelial cell interactions leads to pathophysiological vascular changes similar to diabetic stimulation. a** Timeline of treatment administration aimed at inhibiting pericyte-EC stability (top). Schematic of targeted pathways (bottom). Created with BioRender.com. APB5, PDGFRβ inhibitor; Tie2i, TIE2 inhibitor; DAPT, γ-secretase inhibitor; 19,20-dihydroxydocosapentaenoic acid (DHDP); advanced glycation end products (AGE). **b** Representative images of iBRB MVNs (UEA I, top) and overlay images with pericytes (PDGFRβ, middle) on D14, following 7 days of treatment. Enlarged regions of overlay images showing pericyte-EC interactions (bottom). Quantification of vascular area, **P = 0.0066 (**c**), pericyte area, *P = 0.0176, ***P = 0.0001 (**d**), pericyte coverage (**e**) and fraction of ghost vessels (**f**) for untreated conditions (blue) and different treatments (red) on D14. n = 29 untreated, n = 34 APB5, n = 35 Tie2i, n = 30 DAPT, n = 27 DHDP and n = 35 AGE treated networks analyzed from n = 3 replicate

channels for vascular area. n = 20 untreated, n = 22 APB5, n = 23 Tie2i, n = 19 DAPT, n = 17 DHDP and n = 22 AGE treated networks analyzed from n = 2 replicate channels for pericyte area and coverage. n = 9 untreated, n = 12 APB5, n = 12 Tie2i, n = 11 DAPT, n = 10 DHDP and n = 13 AGE treated networks analyzed from n = 1 channel for ghost vessel fraction. Data are mean ± s.d. ****P < 0.0001; one-way ANOVA. Source data are provided as a Source Data file. **g**, Enlarged image regions of endothelial surface stainings (UEA I) appearing disrupted with diabetic (middle) and APB5 (right) treatments compared to the untreated control (left). Arrows indicate granular objects detected on the vascular luminal side. **h**, Appearance of holes in the endothelial surface visible with UEA I (left) and CLDN5 (right) stainings. This qualitative change is specific to APB5 treatment. All images show maximum intensity projections of 395 µm Z-stacks, from n = 2 independent experiments. Scale bars, 20 µm (**b** bottom, **g**, **h**) and 100 µm (b top and middle).

accumulation in DR can lead to pericyte apoptosis via activation of the NF-kB signaling[59]. The compounds were applied to the iBRB-on-a-chip between D7 and D28 and morphological alterations were quantified using the same analysis workflows as described previously. The vascular area decreased with APB5 and Tie2i treatments compared to untreated controls and increased with DAPT (Fig. 5b, c; Supplementary Figs. 16 and 17). Interestingly, visible changes in vessel diameter were obtained with Tie2i treatment causing thinner vessels (10-50 µm diameter) that were strongly CLDN5+, while DAPT caused wider vessels (30–85 µm diameter) with discontinuous CLDN5+ signal (Fig. 5b and Supplementary Fig. 16). Next, the pericyte area and coverage decreased with APB5 treatment compared to the untreated control (Fig. 5d, e; Supplementary Figs. 16 and 17), showing that PDGFRβ inhibition (APB5) led to most pericyte loss. Furthermore, most treatments increased the fraction of ghost vessels compared to the untreated control, similar to diabetic conditions, while DAPT

treatment decreased ghost vessels (Fig. 5f; Supplementary Fig. 16 and 17). Phenotypes on D28 showed further progression compared to D14, resulting in stronger regressive changes for APB5, Tie2i and DHDP treatments, and proliferative changes for DAPT and AGE (Supplementary Figs. 17 and 18). Then, immunostainings revealed new qualitative changes. Firstly, UEA I staining appeared granular with APB5 (Fig. 5g). UEA I is a lectin that binds glycoconjugates presented on human ECs and uneven staining might indicate possible glycocalyx degradation[60]. Granular UEA I staining was also observed with IL-6 treatment, which might indicate that IL-6 is the causative factor for glycocalyx damage in the diabetic cocktail, since hyperglycemia or TNF-α treatments alone did not alter UEA I (Supplementary Fig. 10). Secondly, UEA I and CLDN5 stainings also revealed the appearance of unstained holes in the microvascular surface with PDGFRβ inhibition, similar but superior to diabetic treatment (Fig. 5h and Supplementary Fig. 16). In conclusion, we observed treatment-specific microvascular

alterations in the iBRB-on-a-chip, linking the inhibition of specific pathways to phenotypic changes.

## Discussion

Chronic microvascular disorders such as DR are poorly understood due to the absence of experimental models that capture comprehensive pathophysiological changes, and there is no treatment available for patients that addresses early stages of the disease. Human model systems can complement animal models and increase the quality of candidate molecules that are brought to the preclinical stage as well as improve confidence in target mechanism of action. Current in vitro models of the iBRB are based on simple, static 2D Transwell inserts and do not mimic iBRB architecture and physiology. In this work, we developed an iBRB-on-a-chip model with human primary retinal cells that recapitulated the cellular organization and function of the native iBRB, and modeled NPDR phenotypes. Tri-cultures of human retinal microvascular ECs, pericytes and astrocytes formed physiological iBRB MVNs[61,62]. In the absence of astrocytes, the vascular diameter and area increased while ECs alone formed MVNs that became unstable and regressed early. Donor variability prevented MVN formation in some EC mono-cultures, but not in tri-cultures, showing that pericytes and astrocytes have a critical supporting role in forming stable MVNs. In addition, tri-cultures showed barrier-specific marker expression, intercellular junction formation and basement membrane production corresponding to physiological iBRB architecture. Upon diabetic stimulation, iBRB MVNs produced clinically relevant NPDR hallmarks that comprehensively captured disease initiation and progression. Expression data evidenced iBRB dysregulation in diabetic conditions, and targeting pericyte-EC communication pathways altered microvascular properties in a treatment-specific manner. Collectively, the iBRB-on-a-chip models critical NPDR pathophysiological changes, supporting the elucidation of pathogenesis and drug discovery. Therapeutic strategies targeting NPDR aim to prevent disease progression and irreversible damage to vision.

The diabetic iBRB-on-a-chip modeled NPDR pathophysiological changes that include vascular regression, pericyte loss, reduced perivascular coverage, increased number of ghost vessels, and inflammation[23,63,64]. Chronic morphological changes were visualized with high content confocal imaging and quantified using custom automated image analyses, and expression profiles were characterized over time. Confocal imaging indicated reduced PDGFRβ+ and Desmin+ pericyte coverage, and RNA-seq downregulation of pericyte markers following 21 days of diabetic treatment, indicating pericyte loss[65]. Transcriptomic analysis indicated changes in iBRB maintenance, consistent with NPDR hallmarks. Among these changes, we targeted pathways involved in pericyte-EC communication, the PDGFRβ, NOTCH and TIE2 signaling pathways, or interfered with pericyte-EC interactions using pro-inflammatory AGE and DHDP signals. We obtained alterations to vascular and pericyte properties, ghost vessel fraction and CLDN5 expression, demonstrating the ability of the iBRB-on-a-chip to model disease trigger-specific responses. In addition, we showed that pericyte elimination with the PDGFRβ inhibitor APB5 disrupted endothelial integrity, showing a direct effect of pericyte loss on vascular damage. TIE2 inhibition surprisingly resulted in microvascular thinning, warranting further investigation[15]. These observations demonstrate that the iBRB-on-a-chip model can be applied to assess the effect of potential therapeutic strategies targeting pericyte-EC interactions, which is also of relevance for microvascular diseases affecting other organs such as the brain and kidney.

The ANG/TIE2 pathway has been already widely explored to develop therapeutic targets normalizing the vasculature in pathological conditions[66]. In mature microvasculature, ANG1 promotes vessel stabilization while combined ANG2 and VEGF induces sprouting. In the absence of VEGF, ANG2 causes pathological changes including vascular regression[67–69]. DR initiation is thought to be caused by an imbalance in ANG1 and ANG2 levels, and increase in ANG2/ANG1 ratio[70]. Our results showed elevated ANG2, reduced ANG1 expression and no VEGF production, which corresponds to vascular regression. Together, diabetic stimulation caused iBRB dysregulation leading to regressive pathological outcomes. The effects of novel therapeutics in patients are sometimes difficult to explain based on clinical findings only. Advanced cellular models such as the iBRB-on-a-chip may inform on mechanisms of action.

Furthermore, cryosections of diabetic ocular tissues showed changes in the basement membrane composition, and ghost vessels were enriched in hyperglycemic conditions as a consequence of pericyte and capillary dropout[71–73]. Here, COL IV+ areas remained constant in all conditions, while ghost vessels increased with vascular regression. A diabetic vasculopathy model reported basement membrane thickening as main readout but did not detect ghost vessels in hyperglycemic conditions[47]. Although we did not observe thickening, we detected differential expression of ECM components and measured accumulation of collagen type IV in diabetic supernatants on D28, corresponding to this DR phenotype.

Besides morphological alterations to the microvascular networks, diabetic stimulation of the iBRB-on-a-chip increased the levels of markers of inflammation, glycocalyx degradation and possible astrogliosis. We observed upregulation of CXCL1, ICAM-1, IL-1β, IL-6 and MCP-1 markers at the protein and gene levels following 14 days of treatment. It has been reported that chronic hyperglycemia can trigger astrocyte reactivity, which induces overexpression of various cytokines (IL-1β, IL-6, IL-8, IFN-γ, TNF-α) via the NF-κB pathway[74]. Also, vascular dysfunction leads to leukocyte recruitment cascade and the inflammation markers ICAM-1, IL-6 and MCP-1 have been detected in the vitreous humor of patients with diabetic macular edema (DME)[75]. Similarly, anti-inflammatory PEDF was decreased in patients with DME[76]. UEA I staining showed notable granularity in diabetic conditions, and when MVNs were treated with APB5 or IL-6. The endothelial glycocalyx was damaged in diabetic patients and glycocalyx volume was reduced in healthy subjects exposed to acute hyperglycemia (300 mg dl$^{-1}$)[77,78]. Moreover, we obtained downregulation of astrocyte markers in diabetic conditions, which corresponds to retinal astrocyte alterations and loss observed in diabetic animals[79,80]. Producing other DR hallmarks such as basement membrane thickening and stiffening, increased barrier permeability, astrocyte reactivity and immune cell invasion are feasible in this model with additional treatments and assays.

The iBRB-on-a-chip incorporates retinal microvascular cell types that are most directly involved in NPDR pathogenesis, and additional cell types could be incorporated such as Müller glia, microglia, immune and neuronal cell types[26]. Human induced pluripotent stem cell (iPSC)-derived ECs and pericytes were shown to form MVNs in vitro[8,81], and could generate reporter or mutant lines[82] informing on cell type or phenotype-specific contributions to DR pathogenesis. Interestingly, studies suggested that fluctuations of high blood glucose levels rather than sustained hyperglycemia cause pericyte-specific toxicity[83], making alternations of diabetic stimulation and starvation a complementary approach to sustained diabetic treatment. While the main purpose of this work was to induce NPDR phenotypes with diabetic stimulation, additional disease triggers and contexts are also applicable to iBRB MVNs[84]. In diabetic patients, vasoregression results in hypoxia, VEGF secretion and neovascularization corresponding to PDR. The microfluidic chip dimensions prevent ischemia, and diabetic treatment did not induce VEGF secretion, indicating that a different disease trigger is required to stimulate neovascularization. To model PDR, iBRB MVNs could either be treated with a diabetic treatment containing VEGF, or could be placed into a hypoxia chamber or treated with DMOG to induce endogenous VEGF production and neovascularization[85]. Furthermore, revascularization processes of pre-existing ghost vessel networks could also be examined in the context

of PDR, age-related macular degeneration (AMD) and cancer[86,87]. In order to strengthen or accelerate pathophysiological changes, retinal microvascular cells or iBRB MVNs could be pre-conditioned with additional disease triggers before applying diabetic stimulation, such as TGF-β1 inducing senescence[88]. These applications will create complementary disease modeling directions branching from the iBRB-on-a-chip and knowledge of disease phenotypes described here, to advance drug discovery for DR and potentially other microvascular disorders.

## Methods

### Cell culture

Original vials of primary Human Retinal Microvascular Endothelial Cells (HRMVECs, Cell Systems, ACBRI 181) were obtained at passage 3 (P3), Human Retinal Pericytes (HRPs, Cell Systems, ACBRI 183) at P3 and Human Retinal Astrocytes (HRAs, ScienCell, 1870) at P1. The list of all cells used in this study is provided in Supplementary Table I. HRMVECs were cultured in T175 flasks coated for 1 h at 37 °C with 1× Attachment Factor Protein (Gibco, S1006100), in EGM-2 MV Microvascular Endothelial Cell Growth Medium-2 BulletKit (EGM-2 MV, Lonza, CC-3202). HRPs and HRAs were cultured in T175 flasks coated for 1 h at room temperature (RT) with 15 μg ml⁻¹ Poly-L-Lysine (PLL, ScienCell, 0403), in Pericyte Medium (PM, ScienCell, 1201) or Astrocyte Medium (AM, ScienCell, 1801), respectively. Cells were harvested with Accutase (Innovative Cell Technologies, AT104). We prepared working stocks by expanding HRMVECs and HRPs to P5 and HRAs to P3, resuspending $1 \times 10^6$ cells ml⁻¹ in STEM-CELLBANKER (Nippon Zenyaku Kogyo, 11897) freezing medium and freezing $5 \times 10^5$ cells per vial. New frozen vials were used for each experiment.

### Vascular network formation

To generate iBRB MVNs on-a-chip, on day 0 we dissociated nearly confluent T175 flasks of HRMVECs, HRPs and HRAs and resuspended them at $36 \times 10^6$ cells ml⁻¹ in EGM2 MV media containing 1% astrocyte growth supplement (EGM2 MV + 1% AGS, ScienCell, 1852) and 4 U ml⁻¹ Thrombin (stock solution at 100 U ml⁻¹ resuspended in distilled water, Sigma-Aldrich, T4648). We then mixed the three cell types at a 1:1:1 cell number ratio unless mentioned otherwise, and mixed this suspension with an equal volume of 6 mg ml⁻¹ Fibrinogen (Sigma-Aldrich, F8630) resuspended in 1× PBS. Next, 10 μl of cell suspension were loaded into the central channel of microfluidic devices (AIM Biotech, idenTx-1 or idenTx-9). The Thrombin and Fibrinogen in the cell suspension crosslink and form a Fibrin gel within a few minutes. After incubating the devices for 15 min at RT, the side channels were filled with 15 μl of EGM2 MV + 1% AGS containing 60 μg ml⁻¹ Human Plasma Fibronectin Solution (HPF, Angio-Proteomie, cAP-42), water reservoirs were filled with 1 or 2 ml distilled water and plates were incubated for 1 h at 37 °C. Finally, EGM2 MV + 1% AGS supplemented with 50 ng ml⁻¹ VEGF (R&D Systems, 293-VE-050/CF) was added to media reservoirs in the devices, 70 μl on one end and 50 μl on the opposite end of side channels to generate a flow replacing the media in contact with the gel channel. On day 2, an additional 3 x 10⁴ HRMVECs were seeded into each side channel by adding 10 μl cell suspension at $1.5 \times 10^6$ cells ml⁻¹ into each side channel, incubating 5 min at RT, then adding 10 μl cell suspension again, inverting the plates and incubating for 1.5 h at 37 °C. On day 4, EGM2 MV + 1% AGS + 50 ng ml⁻¹ VEGF media required for vessel formation was changed to EGM2 MV + 1% AGS media without additional VEGF, to induce barrier maturation. Media was changed daily by adding 70 μl on one end of the side channel and 50 μl on the opposite end. MVNs were processed on day 7, 14 or 28 for further analysis.

### Treatments

Diabetic media composed of EGM2 MV + 1% AGS supplemented with 30 mM D-glucose solution (Gibco, A2494001), 1 ng ml⁻¹ TNF-α (Novus Biologicals, 210-TA-005/CF) and 1 ng m⁻¹ IL-6 (Invitrogen, A42540) was

applied from day 7 to day 14 or 28 to induce diabetic phenotypes. Diabetic media was newly prepared every 2 or 3 days and changed daily. EGM2 MV + 1% AGS supplemented with 30 mM D-Mannitol (Toronto Research Chemicals, TRC-M165000) in 1× PBS was used as an osmotic control, and EGM2 MV + 1% AGS only as an untreated control. Other treatments were prepared by supplementing EGM2 MV + 1% AGS with 10 μg ml⁻¹ CD140b (PDGFRB) Monoclonal Antibody (APB5, PDGFRβ inhibitor, eBioscience, 14-1402-82), 10 μM Imatinib Mesylate (STI571, PDGFRβ inhibitor, ApexBio, A1805), 10 μM TIE2 kinase inhibitor (ApexBio, A5979), 10 μM DAPT (γ-secretase inhibitor, Tocris, 2634), 30 μM 19,20-dihydroxydocosapentaenoic acid (DHDP) or 500 μg ml⁻¹ Advanced Glycation End product-BSA (AGE, Sigma-Aldrich, 121800-10MG). These treatments were applied from day 7 to 28 to alter pericyte-EC interactions, and fresh media were prepared every 2 days and changed daily.

### Immunostaining

MVNs were fixed with 4% PFA (Thermo Scientific, 11490570) for 1 h at RT, permeabilized with 0.1% Triton X-100 (Sigma-Aldrich, T8787) for 10 min at RT and blocked with 10% Donkey Serum (PAN-Biotech, PANP30-0101). MVNs were then stained with primary antibodies diluted 1:100-1:200 in blocking buffer, overnight at 4 °C. We used anti-CD31 (Abcam, ab24590), anti-PDGFRβ (Abcam, ab32570), anti-S100b (Sigma, S2532), anti-desmin (ab15200), anti-GFAP (Invitrogen, MA5-12023), anti-CLDN5 (Invitrogen, 341600), anti-ZO-1 (Thermo Fisher, 339100), anti-VE-cadherin (Abcam, ab33168), anti-laminin (Abcam, ab7463) and anti-collagen type IV (Sigma, MAB1910) antibodies. Devices were then washed 5 times with 1× PBS for >5 min, and stained with corresponding secondary antibodies Alexa Fluor 488 donkey anti-rabbit (Invitrogen, A-21206) or Alexa Fluor 647 donkey anti-mouse (Invitrogen, A-31571) diluted 1:250 in blocking buffer, overnight at 4 °C. Devices were washed again 5 times with 1X PBS for >5 min, stained with lectin *Ulex europaeus* agglutinin I (UEA I) Rhodamine (Vector Laboratories, RL-1062-2) or phalloidin (Invitrogen, R37112) and DAPI (Invitrogen, R37606), and washed overnight at 4 °C.

### Confocal microscopy

MVNs were imaged on an Opera Phenix Plus High-Content Screening System (PerkinElmer), using the Harmony software (v5.1). We acquired image planes of whole-channel regions in the devices with 5× magnification (pre-scan), determined the central position of channels based on the auto-fluorescence of posts at the gel-media interface in the DAPI channel, and acquired image stacks with 10×, 20× or 40× magnification along the whole channels (rescan). For immunostainings, we set the Alexa Fluor 488 channel at 488 nm excitation and 500−550 nm emission length, Rhodamine at 561 nm and 570−630 nm, Alexa Fluor 647 at 640 nm and 650−760 nm, and DAPI at 405 nm and 435−480 nm. Then, we generally defined pre-scan channels for Alexa Fluor 488 at 100 ms exposure time and −70 μm z-position, Rhodamine at 100 ms and −110 μm, Alexa Fluor 647 at 100 ms and −110 μm, and DAPI at 200 ms and −150 μm for the 10× air and 20× water objectives. For the 40× water objective, we defined Alexa Fluor 488 at 200 ms exposure time and −70 μm z-position, Rhodamine at 200 ms and −110 μm, Alexa Fluor 647 at 200 ms and −110 μm, and DAPI at 300 ms and −150 μm. Rescans acquired image stacks using the same channel settings, starting at −80 μm z-position for 45 slices with 7.4 μm step size for the 10× air objective, −150 μm z-position, 80 slices, 5 μm step size for the 20x water objective, and -50 μm z-position, 250 slices, 0.5 μm step size for the 40× water objective. For perfusion or permeability assays, we set the FITC channel at 488 nm excitation and 500−550 nm emission length, and TRITC at 561 nm and 570−630 nm. Pre-scan channels were set for FITC at 160 ms exposure time and −110 μm z-position, TRITC at 200 ms and −110 μm, and DAPI at 300 ms and −130 μm. Rescans acquired image stacks 4 times at 5 min intervals (t = 0, 5, 10 and 15 min) using the same FITC and TRITC channels, no DAPI, starting at −110 μm

z-position for 30 slices with 10 µm step size, using the 10× air objective. Later, image stacks in 10× (-30 µm z-position, 38 slices, 7.5 µm step size, or 20 slices, 10 µm step size for perfusion) and 20× (−20 µm z-position, 58 slices, 5 µm step size) were also acquired based on fixed plate and device dimensions, and fixed ROI positions in whole channels. Laser power was set to 100% for all image acquisitions. Then, we generated global image MIPs to visualize microvascular properties in whole channels, and field image MIPs for image analysis and representative images.

### Image analysis

In order to account for morphological heterogeneity in one image field to the next, we systematically imaged whole gel channels. Each whole channel included up to 15 regions of interest (ROI) per channel for 20× magnification images, which were used for all quantifications. In each experiment, we quantified 3 or more whole channels per condition that we considered as technical replicates, and repeated conditions in 3 independent experiments.

Analysis of MIPs was performed in the Harmony software (v5.1). For all channels, we set intensity values between 0 and 20000, applied median smoothing of 3 px, and then defined channel-specific image regions with common thresholding of 0.25 for ECs (UEA I), 0.40 for pericytes (PDGFRβ) and 0.40 for astrocytes (S100b). Using the DAPI channel, we defined nuclei based on a segmentation algorithm and morphological properties. Similarly, the ECM (COL IV) region was defined by applying median smoothing of 10 px and a common threshold of 0.40. Using these output regions and populations, we defined nuclei in each channel-specific image region by selecting the nuclei overlapping the EC (UEA I) region plus an outer border of 3 µm, the pericyte (PDGFRβ) region plus 2 µm, the astrocyte (S100b) region plus 2 µm, and the ECM (COL IV) region plus 3 µm. Finally, we quantified the total area for all regions and number of objects for nuclei. For pericyte coverage, we defined the pericyte (PDGFRβ) region overlapping with the EC (UEA I) region plus an outer border of 5 µm, and quantified the resulting coverage area. For the ghost vessel fraction, we subtracted the EC (UEA I) vascular area to the ECM (COL IV) area to obtain the avascular COL IV area, and then divided it by the ECM (COL IV) area to obtain the fraction of avascular COL IV, or fraction of ghost vessels.

The Opera Phenix tif image stacks were analyzed with custom python scripts in a two-step approach first using thresholding methods and then training a random forest algorithm on manually improved segmented images from the thresholding approach. Using the scikit-image library[89], a script was developed to segment nuclei, astrocytes and pericytes with multi-otsu thresholding with 3 classes after smoothing and applying a rolling ball algorithm to denoise the images. Identified objects were further processed with closing and opening operations and fused objects were separated with a watershed algorithm. The vasculature was segmented with the random walk algorithm after denoising with Gaussian smoothing and background subtraction with rolling ball. Upper and lower bounds for the random walk algorithm were again determined with 3 class multi-otsu. The thresholding analysis was applied to a subset of images acquired on different days. The generated segmentation masks were inspected with napari and corrected where required. The improved masks were then used to train a random forest algorithm (scikit-learn package) to segment unseen images[90]. The segmentation masks for nuclei, astrocytes and pericytes were again separated with watershed.

To estimate the proximity of astrocytes and pericytes to vasculature, all objects were dilated by 3 pixels in x, y and z directions. The overlapping volume was then measured as an indicator of proximity. The distances between objects were calculated by first estimating surfaces with the marching cube algorithm and applying a k-dimensional tree for partitioning space. The shortest distance between each pericyte or astrocyte and the closest vessel was measured in the x, y and z-directions, and averaged for each ROI.

### Cell viability assays

CellEvent™ Caspase-3/7 Detection Reagents (Invitrogen, C10423) were applied to image apoptotic nuclei and detect toxicity in short- and long-term treatment conditions on D7, D9, D11, D14 and D28. Positive controls were pre-treated for 6 h with 1 µM Staurosporine, an apoptosis inducer, in EGM2 MV + 1% AGS tri-culture media. Then, 1× staining solution at 5 µM in tri-culture media was applied to the reservoirs of the microfluidic devices, and incubated for 60 min at 37 °C. Afterwards, cells were fixed, co-stained with UEA I, PDGFRβ and DAPI to determine co-localization of apoptotic nuclei with specific cell types.

LDH-Glo™ Cytotoxicity Assay (Promega, J2380) was used to detect LDH release in supernatants collected on D7, D9, D11, D14 and D28. Positive controls for maximum LDH release were treated with 0.9% Triton X-100 on D7 and D28. Supernatants were diluted 1:100 in LDH storage buffer (200 mM Tris-HCl (pH 7.3), 10% Glycerol, 1% BSA) and stored at -20 °C. These samples were then mixed 1:1 with LDH detection reagent and luminescence was measured after 60 min incubation at RT.

### Permeability assay

To quantify permeability, we perfused MVNs with 70 kDa TRITC-dextran (100 µg ml⁻¹ in EGM2 MV + 1% AGS tri-culture media) by applying interstitial pressure. Then, we acquired image stacks with confocal microscopy at t = 0, 5, 10 and 15 min following addition of the dextran solution, generated MIPs, prepared binary images, measured fluorescent areas, and calculated permeability coefficients as described previously[13,14]. The permeability was measured on D10. As a positive control, we treated MVNs with 1 ng ml⁻¹ TNF-α for 24 h to induce barrier breakdown and permeability increase.

In whole channels, perfusability was first measured by overlaying global 20× UEA I staining images onto 10× TRITC perfusion images, and quantifying the percentage of perfused area compared to total vessel area on D28 in the overlapping region. Images from each stack were stitched to form global images, and maximum z-projections were prepared and thresholded. To quantify permeability coefficients, ROIs of perfused MVNs were selected and the TRITC intensity inside and outside the vessel areas measured at t = 5 and t = 15 min on D14 and D28. Image processing and analyses were performed using NIH ImageJ software[91].

### Differential gene expression and gene-set enrichment analysis

Total RNA was isolated at relevant time points, pooled from three replicate devices per condition, and stored at −80 °C until analysis. For this, we manually delaminated the membrane sealing on the bottom surface of the devices to expose microfluidic channels. We purified total RNA using an RNeasy Micro Kit (Qiagen) starting by lysing and collecting iBRB MVNs in the gel channel with lysis buffer, and then following the kit protocol. Sample RNA sequencing and primary analysis was outsourced to Microsynth. In brief, library preparation and sequencing included quality control of total RNA samples, preparation of Illumina stranded, poly-A enriched TruSeq RNA libraries, sequencing on Illumina NovaSeq, S2, 2x100bp, and demultiplexing and trimming of Illumina adaptor residuals. Bioinformatic analysis consisted of differential gene expression analysis (GRCh38 ENSEMBL) and pathway analysis[92]. Top depleted and enriched KEGG pathways[93] compared treatment conditions and time points. We used KNIME to extract Log2 FC for defined gene panels. Finally, we performed pathway analysis with Metascape (https://metascape.org), using as input genes with | Log2 FC| > 1, and output the top 20 up- and down-regulated Reactome pathways in diabetic conditions on D14 and D28[94,95].

## Luminex assay

Supernatants were collected at relevant time points, pooled from three to nine replicate devices per condition, and stored at −80 °C until analysis. Measurements of the cytokine concentrations in the supernatant samples were performed using three different customized configured Luminex Human Discovery Assays (8-, 16- and 18plex from R&D Systems) according to the manufacturer's instructions. In order for the samples to be in the linear region of the standard curve, they had to be diluted 1:5 or 1:200 depending on the analyte. Quantification was performed using a Luminex Flexmap 3D device and analysis of protein concentration by standard curve interpolation performed using the xPonent software (v4.2).

## Statistical analysis

All data are plotted as mean ± s.d. Statistical significance was assessed using one-way analysis of variance (ANOVA) followed by Šidák's multiple comparisons test with the software GraphPad Prism (v8.4.2). Significance is represented as ns for not significant, $*P < 0.05$; $**P < 0.01$; $***P < 0.001$; $****P < 0.0001$. In each experiment, at least three devices were used per condition unless stated otherwise (see figure legends for detailed information).

## Reporting summary

Further information on research design is available in the Nature Portfolio Reporting Summary linked to this article.

## Data availability

The transcriptomic data set is available through GEO under accession number "GSE228900". All raw data needed for quantification are available in the source data files. Any additional requests for information can be directed to, and will be fulfilled by, the corresponding authors. Source data are provided with this paper.

## Code availability

Image analysis sequences generated in Harmony, the Python script used for 3D image analysis and ImageJ macros for global image alignment and permeability quantifications are available on request from the corresponding author.

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

## Acknowledgements

T.L.M. was supported by a Roche Postdoctoral Fellowship (2020–2023, RPF-ID:565) and R.D.K. received support from the US National Institutes of Health (grant number 5R01NS121078). We thank Denisa Jurčíková, Surayo Akramhanova and Cheyenne Diem for constructive discussions, and Claudia Korn for feedback on the manuscript. Schematics were created with BioRender.com.

## Author contributions

T.L.M. and H.R. conceived the study and designed the experiments; T.L.M. performed the experiments and analyzed the data; A.J.S. and J.R.V. analyzed perfusability and permeability assays, G.S. performed Luminex assays; M.B. developed 3D image analysis methods; S.Y.L. performed 63x confocal microscopy; G.P. provided technical support; T.L.M. and H.R. interpreted the data; H.R., R.D.K., P.D.W. and S.F. provided supervision; T.L.M. and H.R. wrote the manuscript with feedback from all authors.

## Competing interests

T.L.M., A.J.S., G.S., M.B., J.R.V., S.Y.L., S.F., P.D.W. and H.R. are employees of F. Hoffmann-La Roche Ltd. R.D.K. is a cofounder of AIM Biotech. G.P. declares no competing interests.
