## [Peer Review File · Nature Communications]

REVIEWER COMMENTS

Reviewer #1 (Remarks to the Author):

Maurissen et al. reported the development of a useful iBRB-on-a-chip model that mimics the diabetic phenotypes of neurovascular units and suggested that this model may provide an additional understanding of iBRB properties and a drug evaluation platform for pericyte-endothelial cell interactions. However, the inability to ensure consistency in the phenotype of the neurovascular unit in the iBRB-on-a-chip model undermined this manuscript.

Major concerns

1. It is imperative to minimize platform variations that reflect phenotypes when utilizing this chip model to study diabetic iBRB. However, it appears that the vascular phenotypes of the iBRB-on-a-chip model developed in this study were affected by different donors. (Extended Data Fig. 1e,f).

2. What are the clinical rationales behind the creation of a co-culture system with HRMVECs, HRPs, and HRAs in a 1:1:1 ratio?

2-1. The Muller cells, one of the important cells that make up the neurovascular unit and play a crucial role in maintaining the inner BRB and vascular leakage, were excluded in this study. What is the reason for this exclusion?

3. Changes in the secretome of various growth factors and inflammatory mediators involved in retinal vascular permeabilization and neurovascular unit damage have been reported in diabetic animal models and patients (Progress in Retinal and Eye Research 63 (2018) 20–68).

What is the rationale for selecting TNF- α and IL-6 to be added with high glucose condition in this model instead of other factors such as VEGF, Ang-2, Angiotensin II, C3a, IL-1 β , IL-8, etc. which are also known to affect junctional integrity alteration, pericyte loss, and astrocyte loss causing inner BRB disruption.

4. Based on the results in Fig. 3 and Fig. 4, the changes in inner BRB phenotypes, including decreased vascular area, pericyte loss, increased ghost vessels, and altered transcriptomes and proteins at D28, were significantly different between the control and diabetic groups compared to D14. The authors only investigated endothelial cell-pericyte interactions upon treatment with various drugs and molecules at D14. Therefore, I suggest conducting additional studies to assess any long-term effect at D28.

5. The discussion overstates the results. Retinal neurovascular unit refers to the interdependency of the endothelial cells with the pericytes, glia, neurons and resident immune cells. An early deficiency in the function of the retinal neurovascular unit in diabetes retinopathy results in impaired neurovascular coupling, loss of autoregulation and microglial activation. It is not rigorous to imitate and model the NPDR phenotype and retinal neurovascular units only by the three cell types that composed of iBRB.

6. Compared with the static cell culture, on-a-chip systems have numerous advantages, for example, physiological structure including ECM, cell-cell interaction and vasculature-like perfusion. But from the results of confocal images, the iBRB-on-a-chip does not represent natural iBRB architecture. Mural cells are constituting a variety of cell phenotypes distributed along the microvascular tree, ensheathing pericyte in larger diameter vessel and mesh pericyte in the capillary bed. The results do not show the phenotypes changes of the pericytes, we question whether the 'vessel maturation or iBRB architecture' is formed and whether cell-cell interactions are occurred. Or what are the hallmarks of vessel maturation'.

Reviewer #2 (Remarks to the Author):

In this manuscript, the authors developed an iBRB model using microfluidics by incorporating triculture model of endothelial cell, astrocyte and pericyte. The strength of this manuscript lies in that the authors were able to successfully reproduce a pathophysiological phenotype and disease pathways in the presented 3D in vitro model and relate it to presenting pericyte-endothelial cell stabilizing strategies and potential targets for diabetic retinopathy. Overall, the manuscript seems to provide technically sound and logical investigation throughout. However, there are some comments to further improve the manuscript:

1. In the introduction, adding some more references on previous research regarding iBRB model would emphasize the novelty of the study and be helpful in explaining the issues regarding iBRB. While the authors mention that there is a lack of research on iBRB, it would be beneficial to mention even in vitro models, such as the transwell version, not just chips. In addition, the authors may even add and relate to recent efforts of developing OBRB model on chips as the usage of microvasculature using vasculogenesis methods are very relevant.

2. The resolution of the figures is very low, making it particularly difficult to discern the graphs and important images. In the case of Fig. 2F, the scale cannot be determined.

3. While the authors explain the method to estimate the proximity of astrocytes and pericytes to vasculature, it is not clear if the method is analyzing the distance in 3D, especially in z direction. How are the distance in z-direction measured?

4. Excessive expansion of the number of data measured in a single chip may exaggerate statistical significance. Specifically, in the case of Fig. 3, several hundred data points were measured in only three

experiments (chips?), and the error bars are quite large. While it is good to measure multiple regions of interest (ROIs) per chip to increase the quality of data, selecting a reasonable minimum area as one ROI for averaging out may be needed such as in Fig. 2F for verifying statistical significance. I am aware that some assert that data averaged from one chip should be counted as one data point when calculating statistical significance, while some others use about 3-5 ROIs per chip with at least 3-5 chips for analysis. Also as one AIM chip consists of 3 independent experimental units, it would be more appropriate to mention what exactly the authors mean by 3 independent experiments.

5. While there is quantification data for the vascular area, there appears to be no measurement data for the diameter. Since the diameter is related to the main role of astrocytes and pericytes, it would be beneficial, adding quantification data to strengthen the argument.

6. The main novelty made in the paper is that iBRB was reproduced using astrocytes and pericytes. Therefore, it appears necessary to verify whether the absence of astrocytes and pericytes actually weakens the barrier function, and whether protein expressions such as ZO-1 and CLDN5 decrease. Currently, due to the absence of a Control group, it seems difficult to claim that Tri-culture actually induced the formation of iBRB

Reviewer #3 (Remarks to the Author):

The study presents an in vitro model system that mimics the physical and biochemical properties of the inner blood-retina barrier (iBRB) using primary human retinal cells. The authors showcased the model's usefulness in studying the mechanisms involved in the development of diabetic retinopathy (DR) by utilizing a diabetic cell culture condition. The iBRB-on-chip presents promising opportunities to enhance our comprehension of the early stages of DR and facilitate the development of novel therapies for this incapacitating complication of diabetes.

1. Animal models, including rodents and primates, have been crucial in preclinical research for understanding the pathogenesis of diabetic retinopathy (DR) and developing new therapeutic strategies. Acknowledging the value of animal models in research and their contributions to medical advances is important, including the development of new treatments for DR.

2. Given that Müller cells play an integral role in the inner blood-retina barrier (iBRB), their inclusion in any in vitro model aiming to replicate the iBRB is crucial. If, due to technical limitations, it is not feasible to include Müller cells in the model system, please acknowledge this limitation to avoid potential misinterpretation of the findings.

3. If applicable, could the authors provide the references for the growing medium conditions used in the study? if the study employs a novel method for the growing medium condition, could you please provide details on the optimization process for each cell type and the tri-culture system used?

4. Could the authors provide a justification for selecting TNF and IL-6 to mimic the diabetic condition in the study? It would be helpful if the authors could comment on the potential impact of excluding pro-inflammatory mediators on the observed effects.
5. Please provide data on cell viability data at days 7, 14, and 28 of the experiment. Additionally, did the diabetic cell culture condition induce more cell death when compared to the control condition?
6. Please provide additional information or references to justify the selection of DHDP and AGE to study pericyte-endothelial cell interactions?
7. Would it be possible to utilize TUNEL or Annexin V assays to identify the initial cell type undergoing apoptosis in the diabetic condition of the tri-culture system? It would be interesting to know if this approach could provide insight into the sequence of events in non-proliferative DR and facilitate the discovery of potential therapeutic targets
8. Could you provide information on previous studies that have utilized on-chip in vitro models with retinal cells to mimic the blood-retinal barriers? Mentioning these studies would help to emphasize the innovative approach of the current study.

Reviewer #4 (Remarks to the Author):

Maurissen et al report an interesting and novel creation of an in vitro model for the blood-retinal barrier (BRB). The authors combine human retinal microvascular cells, pericyte, and astrocytes in a fibrin gel with microfluidics to extremely well characterize a microvascular endothelial network that includes lumens, tight junctions, pericyte coverage, and low permeability. Next, the authors create a diabetes model, including high glucose, IL6, and TNFa, and show that diabetes to capillary dropout, pericyte loss, and ghost vessels in a thorough manner. These alterations are accompanied by gene and protein express changes that mimic what is known about gene expression alterations in early NPDR. Finally, the authors disrupt EC-pericyte interactions through multiple mechanisms and recapitulate aspects of early NPDR reported in vivo. The manuscript is novel, well-written, and very thorough. My major concern is the very high non-physiologic IL6/TNFa concentrations used to create the diabetes model.

Major Comments:

1. IL6 levels in aqueous humor and vitreous humor have been previously reported. In the aqueous humor (PMID: 22511846), IL6 levels range from 1.9-16.4 pg/ml in control and DR eyes while TNFa levels were 0-7.6 pg/ml. In the vitreous humor (PMID: 19997642), IL6 levels range from 12-330 pg/ml in control and PDR eyes, while TNFa levels were not detectable. The authors use 1 ng/ml IL6 and TNFa in their diabetes model, which are greater than physiologic or pathophysiologic levels. In extended Fig 10, it is clear that IL6 alone can reduce pericyte coverage, where TNFa alone has little effect. Based upon this

knowledge and extended Fig 10, I am concerned that the “diabetes model” might be more of a model of supraphysiologic IL6-induced pericyte toxicity, which would significantly hamper the conclusions of this study.

2. In figure 2e, the authors measure permeability of their microvascular network using TRITC-labeled 70 kDa dextran, which is very cool. The authors show that TNFa increases permeability in the binary mask. From the original image, it is difficult for me to appreciate the leakage shown in the binary mask. Can this image quality be improved so the leakage is apparent? Can we show a control for comparison? Why wasn't this important assay performed in the “diabetes” model?

3. The authors state that, “Addition of astrocytes reduced the vascular area and diameter, making the networks more microvascular-like” – I do not see any data on diameter, can this be quantified like vascular area?

Minor Concerns:

1. The authors state: “HRMVECs isolated from different donors did not all lead to the formation of MVNs” in the results, and “Original vials of primary Human Retinal Microvascular Endothelial Cells (HRMVEC, Cell Systems, ACBRI 181) were obtained at passage 3 (P3), Human Retinal Pericytes (HRP, Cell Systems, ACBRI 183) at P3 and Human Retinal Astrocytes (HRA, ScienCell, 1870) at P1” in the methods. Does this mean that the authors purchased cells from different donors and compared them? I don't see any mention of this possibility on their website and want to make sure that I, and the future reader, understand correctly. Is this merely cell quality?

REVIEWER COMMENTS

Reviewer #1 (Remarks to the Author):

Maurissen et al. reported the development of a useful iBRB-on-a-chip model that mimics the diabetic phenotypes of neurovascular units and suggested that this model may provide an additional understanding of iBRB properties and a drug evaluation platform for pericyte-endothelial cell interactions. However, the inability to ensure consistency in the phenotype of the neurovascular unit in the iBRB-on-a-chip model undermined this manuscript.

We thank the reviewer for their careful reading and helpful comments on the manuscript. We appreciate that the reviewer acknowledged that we developed a ‘useful iBRB-on-a-chip model’. We apologize that we did not make the robustness and consistency of the model and phenotypes more clear in the original version of the manuscript. We have now made this point more central in the revised version of the manuscript. Further, in responding to this reviewer’s comments, we have substantially clarified how this work advances beyond established models of retinal microvasculature. We have responded to all of the reviewer’s comments, detailed in the point-by-point response below, and believe that the manuscript has been strengthened substantially by including the additional experiments and clarifications that were suggested.

Major concerns

1. It is imperative to minimize platform variations that reflect phenotypes when utilizing this chip model to study diabetic iBRB. However, it appears that the vascular phenotypes of the iBRB-on-a-chip model developed in this study were affected by different donors. (Extended Data Fig. 1e,f).

We agree that minimizing platform variations and ensuring reproducibility of the vascular network for different donors and preparations is a key feature of any robust organ-on-a-chip model. In general, donor variability is a common limitation for cellular assays. There are often phenotypic differences between primary cell lots that can be attributed to donor different genomic backgrounds, cell purification methods, cell expansion and maintenance, and quality control. Even in induced pluripotent stem cell (iPSC) lines, there can be phenotypic differences between donors, and even between clones obtained from the same donor.

In the context of an iBRB model, primary retinal cells are a limited resource: 1) these cells can only be obtained from deceased donors, 2) few cell lots are available from vendors, and 3) donor information is limited, making it difficult to interpret factors that might affect vascular phenotypes. Nonetheless, per definition, human primary retinal microvascular cells are most relevant to generate an iBRB and model diabetic retinopathy phenotypes.

To ensure robustness of our model, we compared primary human retinal microvascular endothelial cells (HRMVECs) from 7 different donors to assess if cell origin and donor affected 3D microvascular network formation in vitro. When HRMVECs were seeded in the

gel alone (mono-culture condition), 4 out of 7 donors formed 3D networks and 3 out of 7 failed to form networks (new Extended Data Fig. 2a). Importantly, when HRMVECs were cultured together with primary human retinal pericytes (HRP) and primary human retinal astrocytes (HRA) in the gel (co-culture condition), all HRMVEC donors formed 3D networks (new Extended Data Fig. 2b). These results highlight that HRMVECs in our tri-culture system formed 3D microvascular networks independent of the donor, and that co-culture with HRP and HRA was critical for 3D network formation in vitro. In addition, we tested HRA from two different donors and one immortalized human cortical astrocyte cell line (HCA). We observed similar responses to the diabetic cocktail with the different astrocyte batches (new Extended Data Fig. 11b).

As we agree that the enhanced robustness across donors is a unique feature of our tri-culture system, we have highlighted this aspect in the manuscript in the Results and Discussion sections.

Page 7 lines 138-141:

“Importantly, when HRMVECs from different donors were co-cultured with HRP and HRA, all HRMVEC donors formed 3D networks (Extended Data Fig. 2). These results highlight that HRMVECs in our tri-culture system formed 3D MVNs independent of the donor.”

Page 14 lines 310-312:

“Donor variability prevented MVN formation in some EC mono-cultures, but not in tri-cultures, showing that pericytes and astrocytes have a critical supporting role in forming stable MVNs.”

2. What are the clinical rationales behind the creation of a co-culture system with HRMVECs, HRPs, and HRAs in a 1:1:1 ratio?

We have designed our model to resemble closely the *in vivo* architecture of the native tissue and we have selected our design parameters based on information available from human physiology. The endothelial cell to pericyte ratio in human CNS is between 1:1 and 3:1, with data from the retina indicating a ratio closer to 1:1 (Díaz-Flores, L. et al. 2009).

Endothelial cell to pericyte to astrocyte ratios used for in vitro vascular models vary. A human BBB model used 3:1:1 ratio (Campisi, M. et al. 2018), human BRB model used 5:1:5 (Bonkowski, D. et al. 2011; Fresta, C. G. et al. 2020), and rat BBB model used 1:1:5 (Al Ahmad, A. et al 2011).

The clinical rationale for a 1:1:1 ratio was to respect the 1:1 endothelial cell to pericyte ratio in human retinae, and add a similar fraction of astrocytes based on the literature. We clarified this point in the manuscript (page 6 lines 120-121):

“We selected a 1:1 endothelial cell to pericyte ratio to be consistent with the ratio found in human retinas³⁸ and we added a similar astrocyte fraction based on the literature^{13,39–41}.”

2-1. The Muller cells, one of the important cells that make up the neurovascular unit and play a crucial role in maintaining the inner BRB and vascular leakage, were excluded in this study.

What is the reason for this exclusion?

Functional models for drug discovery based on organ-on-a-chip technology must balance complexity in biology and design with ease of use and defined readouts to be relevant for pharmaceutical development. The central aim of our work was to engineer a functional and robust model that recapitulates diabetic retinopathy pathogenesis in a system that is ready for use in the drug development pipeline. In this context, we applied a bottom-up approach. In addition to endothelial cells, we included pericytes as this cell type is present in high abundance in the retina, contributes to iBRB architecture and function, and pericyte pathological phenotypes are observed early in the development of diabetic retinopathy pathogenesis. Astrocytes were included as well as they are known to induce barrier formation in the CNS and retina, promote new vessel stabilization and maturation, and influence vascular morphology in vivo. (Yao, H. et al. 2014; Klaassen, I. et al 2013). In vitro, astrocytes support the formation of functional 3D microvascular networks and increase tight junction expression in endothelial cells (Campisi, M. et al. 2018; Gardner, T. W. et al. 1997).

While Müller cells have a similar role in the maintenance of the BRB as astrocytes, we decided not to include them in this work as it would further complicate the design and use of the model at this stage of development. In addition, at the time of this study, human primary Müller cells were not readily available from suppliers and the immortalized cell line MIO-M1. This model of the inner BRB recapitulates BRB morphology with the formation of vascular networks that are lumenized, including close associations between pericytes and endothelial cells, the deposition of a basement membrane and interactions with astrocytes. Further developments of the iBRB-on-a-chip model could include incorporation of additional retinal cell types that interact directly or indirectly with the inner microvasculature, such as Müller cells, microglia, immune cell types and/or neuronal cell types.

We clarified this point page 17 lines 381-383 of the revised manuscript and have added the following sentence in the Discussion:

“The iBRB-on-a-chip incorporates retinal microvascular cell types that are most directly involved in NPDR pathogenesis, and additional cell types could be incorporated such as Müller glia, microglia, immune and neuronal cell types.”

3. Changes in the secretome of various growth factors and inflammatory mediators involved in retinal vascular permeabilization and neurovascular unit damage have been reported in diabetic animal models and patients (Progress in Retinal and Eye Research 63 (2018) 20–68).

What is the rationale for selecting TNF- α and IL-6 to be added with high glucose condition in this model instead of other factors such as VEGF, Ang-2, Angiotensin II, C3a, IL-1 β , IL-8, etc. which are also known to affect junctional integrity alteration, pericyte loss, and astrocyte loss causing inner BRB disruption.

We thank the reviewer for bringing to our attention this important reference that describes biological factors upregulated in the ocular media of patients with diabetic macular edema and we have added this reference at page 17 line 390 in the revised manuscript (*Daruich, A. et al. 2018*). We have added TNF- α and IL-6 in the diabetic medium as they are important mediators of inflammation in the early phases of DR and are elevated in patients with DR (*Demircan, N. et al 2006; H. Atli, H et al 2022*).

Also, a similar medium composition was investigated in an *in vitro* model of vascular organoids to induce features of diabetic vasculopathy (*Wimmer, R. A. et al. 2019*). The authors used 75 mM glucose combined with 1ng/ml TNF- α and 1ng/ml IL-6 and observed clinically relevant phenotypes in the organoids, such as basement membrane thickening. In our model, we applied the same concentrations of TNF- α and IL-6 as described in the article (1ng/ml), and reduced the glucose concentration to 30 mM to be comparable to other *in vitro* studies on human retinal microvascular endothelial cells (*Rochfort, K. D. et al. 2019; Eyre, J. J. et al 2020*) and to stay within physiological hyperglycemia concentrations. Using these treatment conditions, we established a disease model of diabetic retinopathy that develops pathophysiological changes resembling clinical hallmarks of early DR.

As a follow up study, we will be investigating the effects of additional disease triggers (including Ang2 and VEGF) and medium compositions on the inner BRB-on-a-chip model in order to provide a comprehensive understanding of how different molecular factors induce disease phenotypes *in vitro*.

We have clarified the choice of the diabetic cocktail composition in the manuscript (page 9 lines 173-176):

“Vitreous and plasma levels of TNF- α and IL-6 are elevated in patients with DR and these pro-inflammatory cytokines were added to the high glucose medium to mimic the inflammatory state observed in diabetes⁴⁸⁻⁵¹.”

4. Based on the results in Fig. 3 and Fig. 4, the changes in inner BRB phenotypes, including decreased vascular area, pericyte loss, increased ghost vessels, and altered transcriptomes and proteins at D28, were significantly different between the control and diabetic groups compared to D14. The authors only investigated endothelial cell-pericyte interactions upon treatment with various drugs and molecules at D14. Therefore, I suggest conducting additional studies to assess any long-term effect at D28.

We thank the reviewer for this comment and have performed additional experiments to assess the effects of the inhibitors at D28. Results on D28 are consistent with D14, showing increased morphological differences (new Extended Data Fig. 17a) and sustained expression changes (new Extended Data Fig. 17b) in responses to inhibitors, confirming that long-term

treatment effects on inner BRB phenotypes occur progressively. Interestingly, while most treatments cause regressive phenotypes and loss of inner BRB integrity, DAPT treatment instead shows proliferative phenotypes, with increased vascular area, pericyte area and coverage, and reduced fraction of ghost vessels. These differences become more evident at the later time point D28, indicating that long-term treatment might be required to assess phenotypes with higher confidence, monitor effects over time and potentially model proliferative phenotypes that rescue or enhance inner BRB integrity.

We have added the conclusions of the results at D28 in the manuscript (page 13 lines 284-286):

“Phenotypes on D28 showed further progression compared to D14, resulting in stronger regressive changes for APB5, Tie2i and DHDP treatments, and proliferative changes for DAPT and AGE that enhanced microvascular properties (Extended Data Fig. 17 and 18).”

Extended Data Fig. 17 | Long-term treatment effects. *a*, Quantification of vascular area, pericyte area and coverage, and fraction ghost vessels for untreated, APB5, Tie2i, DAPT, DHDP and AGE-treated conditions on D14 and D28. $n = 45$ untreated D14, $n = 44$ D28, $n = 43$ APB5 D14, $n = 41$ D28, $n = 44$ Tie2i D14, $n = 45$ D28, $n = 44$ DAPT D14, $n = 45$

*D28, n = 44 DHDP D14, n = 45 D28, n = 45 AGE D14 and n = 45 D28 treated networks analyzed from n = 3 replicate channels. b, Analyte measurements of human collagen IV alpha I, IL-6, Angiopoietin-1 and Angiopoietin-2 obtained from n = 3 replicate channels. Data are mean ± s.d. *P<0.05; **P<0.01; ***P<0.001; ****P<0.0001; one-way ANOVA. c, Representative images of endothelial networks (UEA I) and overlay images with pericytes (PDGFRβ) or basement membranes (COL IV) on D28. All images show maximum intensity projections of 245 μm Z-stacks. Scale bars, 100 μm (c).*

5. The discussion overstates the results. Retinal neurovascular unit refers to the interdependency of the endothelial cells with the pericytes, glia, neurons and resident immune cells. An early deficiency in the function of the retinal neurovascular unit in diabetes retinopathy results in impaired neurovascular coupling, loss of autoregulation and microglial activation. It is not rigorous to imitate and model the NPDR phenotype and retinal neurovascular units only by the three cell types that composed of iBRB.

We thank the reviewer for this clarification. We have removed the term neurovascular unit from the discussion (page 14 line 308). This *in vitro* model revolves around the three main cell types forming the BRB, with the possibility to add additional cell types in future studies.

6. Compared with the static cell culture, on-a-chip systems have numerous advantages, for example, physiological structure including ECM, cell-cell interaction and vasculature-like perfusion. But from the results of confocal images, the iBRB-on-a-chip does not represent natural iBRB architecture. Mural cells are constituting a variety of cell phenotypes distributed along the microvascular tree, ensheathing pericyte in larger diameter vessel and mesh pericyte in the capillary bed. The results do not show the phenotypes changes of the pericytes, we question whether the 'vessel maturation or iBRB architecture' is formed and whether cell-cell interactions are occurred. Or what are the hallmarks of vessel maturation'.

We have added additional data to demonstrate the formation of an iBRB architecture in our model.

First, we have taken confocal images of the inner BRB-on-a-chip at D7 at high magnification (63x) to confirm the close interactions between endothelial cells and pericytes in the model (new Extended Data Fig. 3b). On these images, we observe pericytes (labeled with PDGFRb in green) lining the vessels (labeled with UEA1 in orange). The pericytes and endothelial cells are in physical association and are only separated by a thin basement membrane, comparable to what is observed *in vivo*. These images confirmed that cells are interacting with one another in our iBRB-on-a-chip model.

In addition, we have compared the expression of genes related to pericyte-endothelial cell interactions and to basement membrane formation at different time points during the iBRB-on-a-chip maturation phase (D0, D4 and D7) (new Extended Data Fig. 3c). Gene expression of receptors implicated in endothelial cell-pericyte signal transmission either via paracrine signaling (Tie2, PDGFRb) or juxtacrine signaling (Notch) were increased between D0 and

D7. We also noticed increased expression of the gene coding for the endothelium junctional protein Claudin 5. Finally, we observed increased expression of genes coding for basement membrane proteins (i.e., Perlecan, Nidogen, Collagen IV and Laminins 211 and 511) between D0 and D7, suggesting that cells are contributing to vessel maturation and stability by actively depositing a basement membrane.

Altogether these data demonstrate that pericytes and endothelial cells are in close contact in the iBRB-on-a-chip and that our model recapitulates vessel maturation characterized by basement membrane deposition and cellular interactions, allowing us to study early stages of diabetic retinopathy related to pericyte-endothelial cell interactions.

The existing in vitro models of the iBRB that integrate endothelial cells, pericytes and astrocytes are based on 2D Transwell inserts that lack an extracellular matrix, flow, and where cells are seeded on a planar artificial surface (*Wisniewska-Kruk, J. et al. 2012; Sánchez-Palencia, D. M., et al 2016; Bryan, B. A. & D'Amore, P. A. 2008*). Our iBRB-on-a-chip represents a significant advance in the field of human model systems and provide a new tool to explore disease mechanisms and barrier biology.

We have added the following sentence in the manuscript (page 7 lines 146-147) and have added references to the Extended Data Fig. 3:

“We observed close physical interactions between endothelial cells and pericytes (Extended Data Fig. 3).”

Extended Data Fig. 3 | Characterization of iBRB MVN formation. *a*, Representative image comparison between 3D mono- and tri-cultures, and tight junction (CLDN5) integrity. Images show maximum intensity projections of 290 μm Z-stacks. Scale bars, 100 μm . *b*, Confocal images at 63x magnification showing pericytes (PDGFR β green) lining the vessels (UEA I orange). *c*, Normalized counts of genes involved in cell-cell interactions and basement membrane composition during the maturation phase of the iBRB-on-a-chip (D0 to D7). Data are from $n = 3$ independent experiments.

Reviewer #2 (Remarks to the Author):

In this manuscript, the authors developed an iBRB model using microfluidics by incorporating triculture model of endothelial cell, astrocyte and pericyte. The strength of this manuscript lies in that the authors were able to successfully reproduce a pathophysiological phenotype and disease pathways in the presented 3D in vitro model and relate it to presenting pericyte-endothelial cell stabilizing strategies and potential targets for diabetic retinopathy. Overall, the manuscript seems to provide technically sound and logical investigation throughout. However, there are some comments to further improve the manuscript:

We thank the reviewer for their careful reading and helpful comments on the manuscript. We appreciate that the reviewer noted that the manuscript ‘provides technical sound and logical investigation throughout’. We believe that the revised version of the manuscript has been substantially improved based on the careful and helpful comments of the reviewer.

1. In the introduction, adding some more references on previous research regarding iBRB model would emphasize the novelty of the study and be helpful in explaining the issues regarding iBRB. While the authors mention that there is a lack of research on iBRB, it would be beneficial to mention even in vitro models, such as the transwell version, not just chips. In addition, the authors may even add and relate to recent efforts of developing oBRB model on chips as the usage of microvasculature using vasculogenesis methods are very relevant.

We thank the reviewer for this comment. We have added a paragraph about the status of the in vitro models of the iBRB as well as development around oBRB models in the Introduction (page 5 lines 89-100).

“Several organ-on-a-chip technologies have been leveraged for ophthalmology applications to model the choroid, the outer BRB, the interface between photoreceptors and RPE, or investigate pericytes-endothelial cell interactions²⁹⁻³³. There is currently no human model system to study pathophysiological changes in cell types of the iBRB. The models of the iBRB that have been developed so far are based on flat Transwell inserts where endothelial cells, pericytes and/or astrocytes are seeded on a polycarbonate membrane or at the bottom of the well³⁴⁻³⁶. While these Transwell iBRB models have highlighted the contribution of astrocytes to endothelial barrier properties, they lack an extracellular matrix, direct physical contact between the cells and a vascular-like architecture. Reconstructing these specialized features of the iBRB in vitro and interrogating them experimentally will enable better understanding of iBRB properties in physiological and disease conditions as well as provide a platform for validating therapeutic approaches.”

The following references related to iBRB and oBRB models have been added to the manuscript:

Haderspeck, J. C., Chuchuy, J., Kustermann, S., Liebau, S. & Loskill, P. Organ-on-a-chip technologies that can transform ophthalmic drug discovery and disease modeling. *Expert Opin. Drug Discov.* 14, 47–57 (2019).

Yeste, J. et al. A compartmentalized microfluidic chip with crisscross microgrooves and electrophysiological electrodes for modeling the blood-retinal barrier. *Lab Chip* 18, 95–105 (2018).

Achberger, K. et al. Merging organoid and organ-on-a-chip technology to generate complex multi-layer tissue models in a human retina-on-a-chip platform. *Elife* 8, 1–26 (2019).

Rogers, M. T. et al. A high-throughput microfluidic bilayer co-culture platform to study endothelial-pericyte interactions. *Sci. Rep.* 11, 1–14 (2021).

Cipriano, M. et al. Human immunocompetent choroid-on-chip: a novel tool for studying ocular effects of biological drugs. *Commun. Biol.* 5, 1–13 (2022).

Wisniewska-Kruk, J. et al. A novel co-culture model of the blood-retinal barrier based on primary retinal endothelial cells, pericytes and astrocytes. *Exp. Eye Res.* 96, 181–190 (2012).

2. The resolution of the figures is very low, making it particularly difficult to discern the graphs and important images. In the case of Fig. 2F, the scale cannot be determined.

We thank the reviewer for this comment and we have improved the size of Fig. 2f.

Fig. 2 | Inner retinal microvasculature indicate mature and functional barrier properties. *a*, Representative images of tight junction (CLDN5 and ZO-1) and adherens junction (VE-cadherin) proteins co-localizing with endothelial networks (UEA I or CD31). *b*, Basement membrane proteins (LAM and COL IV) co-localizing with endothelial networks (UEA I). Images show maximum intensity projections of 290 μm Z-

stacks. Stainings were repeated in $n = 3$ independent experiments. **c**, Heat map of differential gene expression between D4 and D0 (vessel formation), D7 and D4 (vessel maturation), and the whole iBRB MVN formation process between D7 and D0. Results are from a defined characterization gene panel without cutoff. Data are RNA-Seq aggregated Log2 FC from $n = 3$ independent experiments. **d**, Permeability assay timeline: iBRB MVNs were formed between D0 and D7, were either kept in standard culture or treated with 1 ng ml^{-1} TNF- α for 24 h before performing the permeability assay. **e**, Representative image of TNF- α -treated (1 ng ml^{-1} , 24 h) iBRB MVNs perfused on D10 with TRITC-labelled dextran ($100 \text{ }\mu\text{g ml}^{-1}$, 70 kDa), acquired by fluorescence confocal microscopy at 5 min intervals. At $t = 10 \text{ min}$, the binary mask shows leakage. Perfusion images are maximum intensity projections of $290 \text{ }\mu\text{m}$ Z-stacks. **f**, Apparent permeability coefficients were quantified on D10 in untreated and TNF- α -treated MVNs using 70 kDa TRITC-dextran as a tracer. $n = 6$ untreated and $n = 3$ TNF- α -treated networks. Data are mean \pm s.d. $**P = 0.0029$; two-tailed Student's t -test. Scale bars, $100 \text{ }\mu\text{m}$ (**a**, **b**, **e**).

3. While the authors explain the method to estimate the proximity of astrocytes and pericytes to vasculature, it is not clear if the method is analyzing the distance in 3D, especially in z direction. How are the distance in z-direction measured?

We have clarified in the Methods section that the distance between cells is also measured in the z-direction (page 24 lines 547-549):

“The shortest distance between each pericyte or astrocyte and the closest vessel was measured in the x, y and z-directions, and averaged for each ROI.”

4. Excessive expansion of the number of data measured in a single chip may exaggerate statistical significance. Specifically, in the case of Fig. 3, several hundred data points were measured in only three experiments (chips?), and the error bars are quite large. While it is good to measure multiple regions of interest (ROIs) per chip to increase the quality of data, selecting a reasonable minimum area as one ROI for averaging out may be needed such as in Fig. 2F for verifying statistical significance. I am aware that some assert that data averaged from one chip should be counted as one data point when calculating statistical significance, while some others use about 3-5 ROIs per chip with at least 3-5 chips for analysis. Also as one AIM chip consists of 3 independent experimental units, it would be more appropriate to mention what exactly the authors mean by 3 independent experiments.

In order to account for morphological heterogeneity in one image field to the next, we systematically imaged whole gel channels. Each whole channel included up to 15 regions of interest (ROI) per channel for 20x magnification images, which were used for all quantifications. In each experiment, we quantified 3 or more whole channels per condition that we considered as technical replicates. In Fig. 3, the quantification was repeated in 3 independent experiments. This would total to approximately $3 \times 3 \times 15 = 135$ single values that we plotted on the figure.

We clarified this point in the Methods section (page 22 lines 510-514):

“In order to account for morphological heterogeneity in one image field to the next, we systematically imaged whole gel channels. Each whole channel included up to 15 regions of interest (ROI) per channel for 20x magnification images, which were used for all quantifications. In each experiment, we quantified 3 or more whole channels per condition that we considered as technical replicates, and repeated conditions in 3 independent experiments.”

In addition we prepared graphs where measurements for each chip were averaged and repeated statistical analysis. We observed the same trends as when the quantifications were performed by field of view. The mean vascular area and pericyte coverage per channel decreased significantly with diabetic treatment compared to respective time-matched controls (panel c). New graphs are referenced as Extended Data Fig. 1e (page 7 line 132), Extended Data Fig. 5b (page 9 line 183), Extended Data Fig. 6c, g (page 9 lines 190 and 195).

Extended Data Fig. 1e (top), Quantification of mean vascular, pericyte and total areas per channel. $n = 3$ HRMVEC 1:0:0, $n = 3$ 1:1:0, $n = 3$ 1:1:1 and $n = 3$ HUVEC 1:1:1 whole channels analyzed from $n = 3$ independent experiments.

Extended Data Fig. 5b (bottom left), Quantification of mean vascular area per channel. $n = 9$ control D7, $n = 9$ D14, $n = 12$ D28, $n = 9$ diabetic D14 and $n = 15$ D28 whole channels analyzed from $n = 3$ independent experiments.

Extended Data Fig. 6c (bottom second and third), Quantification of mean pericyte area and coverage per channel. $n = 4$ control D7, $n = 4$ D14, $n = 4$ D28, $n = 4$ diabetic D14 and $n = 5$ D28 whole channels analyzed from $n = 3$ independent experiments.

Extended Data Fig. 6g (bottom fourth and fifth), Quantification of mean COL IV area and avascular area per channel. $n = 3$ control D7, $n = 3$ D14, $n = 3$ D28, $n = 3$ diabetic D14 and $n = 3$ D28 whole channels analyzed from $n = 3$ independent experiments.

5. While there is quantification data for the vascular area, there appears to be no measurement data for the diameter. Since the diameter is related to the main role of astrocytes and pericytes, it would be beneficial, adding quantification data to strengthen the argument.

We agree with the reviewer and have quantified the diameter of the vascular networks in the monoculture and tri-culture conditions. On the left figure, the mean diameter per channel was quantified by dividing the total vascular area by the total skeleton length on 3 replicate channels from independent experiments. The figure on the right displays the distribution of vessel diameter within one channel quantified manually. Both image analyses show the vessel diameter was reduced when astrocytes and pericytes were added to the endothelial cells. The figures have been included in Extended Data Fig. 1f, g (page 7 line 133).

6. The main novelty made in the paper is that iBRB was reproduced using astrocytes and pericytes. Therefore, it appears necessary to verify whether the absence of astrocytes and pericytes actually weakens the barrier function, and whether protein expressions such as ZO-1 and CLDN5 decrease. Currently, due to the absence of a Control group, it seems difficult to claim that Tri-culture actually induced the formation of iBRB

We agree with the reviewer that adding astrocytes and pericytes is a strength of the model and have made more clear that the tri-culture substantially improves vascular formation as compared with mono-culture. We have provided additional images of mono-culture (HRMVEC) and tri-culture (HRMVEC:HRP:HRA) stained with UEA1 and Claudin-5 (new Extended Data Fig 3a). These images demonstrate the formation of a vascular network in the tri-culture condition, while networks were defective in mono-culture condition. Vascular networks composed of endothelial cells only did not remain stable over time, emphasizing the critical importance of pericytes and astrocytes to the formation and maintenance of the iBRB model.

We have clarified these points in the Manuscript:

Page 7 lines 138-141:

“Importantly, when HRMVECs from different donors were co-cultured with HRP and HRA, all HRMVEC donors formed 3D networks (Extended Data Fig. 2). These results highlight that HRMVECs in our tri-culture system formed 3D MVNs independent of the donor.”

Page 14 lines 310-312:

“Donor variability prevented MVN formation in some EC mono-cultures, but not in tri-cultures, showing that pericytes and astrocytes have a critical supporting role in forming stable MVNs.”

Reviewer #3 (Remarks to the Author):

The study presents an in vitro model system that mimics the physical and biochemical properties of the inner blood-retina barrier (iBRB) using primary human retinal cells. The authors showcased the model's usefulness in studying the mechanisms involved in the development of diabetic retinopathy (DR) by utilizing a diabetic cell culture condition. The iBRB-on-chip presents promising opportunities to enhance our comprehension of the early stages of DR and facilitate the development of novel therapies for this incapacitating complication of diabetes.

We thank the reviewer for their careful reading of the manuscript and helpful comments. We appreciate that the reviewer thinks our model will ‘enhance our comprehension’ of the disease and ‘facilitate the development of novel therapies’. To strengthen this aspect, we have performed additional experiments where we tested the components of the diabetic cocktail individually and performed additional viability assays.

1. Animal models, including rodents and primates, have been crucial in preclinical research for understanding the pathogenesis of diabetic retinopathy (DR) and developing new therapeutic strategies. Acknowledging the value of animal models in research and their contributions to medical advances is important, including the development of new treatments for DR.

We agree with the reviewer that animal models are highly valuable in research and we believe that human model systems are not meant to replace in vivo models but are complementary. We have added the following sentences in the Discussion (page 14 lines 299-307).

“Chronic microvascular disorders such as DR are poorly understood due to the absence of experimental models that capture comprehensive pathophysiological changes, and there is no treatment available for patients that address early stages of the disease. Human model systems can complement animal models and increase the quality of candidate molecules that are brought to the preclinical stage as well as improve confidence in target mechanism of action. Current in vitro models of the iBRB are based on simple, static 2D Transwell inserts and do not mimic iBRB architecture and physiology. In this work, we developed an iBRB-on-a-chip model with human primary retinal cells that recapitulated the cellular organization and function of the native iBRB, and modeled NPDR phenotypes.”

2. Given that Müller cells play an integral role in the inner blood-retina barrier (iBRB), their inclusion in any in vitro model aiming to replicate the iBRB is crucial. If, due to technical limitations, it is not feasible to include Müller cells in the model system, please acknowledge this limitation to avoid potential misinterpretation of the findings.

We agree with this point which was also raised by Reviewer 1 Comment 2.1. Functional models for drug discovery based on organ-on-a-chip technology must balance complexity in biology and design with ease of use and defined readouts to be relevant for pharmaceutical development. The central aim of our work was to engineer a functional and robust model that recapitulates diabetic retinopathy pathogenesis in a system that is ready for use in the drug development pipeline. In this context, we applied a bottom-up approach. In addition to endothelial cells, we included pericytes as this cell type is present in high abundance in the retina, contributes to iBRB architecture and function, and pericyte pathological phenotypes are observed early in the development of diabetic retinopathy pathogenesis. Astrocytes were included as well as they are known to induce barrier formation in the CNS and retina, promote new vessel stabilization and maturation, and influence vascular morphology in vivo. (Yao, H. et al. 2014; Klaassen, I., et al 2013). In vitro, astrocytes support the formation of functional 3D microvascular networks and increase tight junction expression in endothelial cells (Campisi, M. et al 2018; Gardner, T. W. et al 1997).

While Müller cells have a similar role in the maintenance of the BRB as astrocytes, we decided not to include them in this work as it would further complicate the design and use of the model at this stage of development. In addition, at the time of this study, human primary

Müller cells were not readily available from suppliers and the immortalized cell line MIO-M1. This model of the inner BRB recapitulates BRB morphology with the formation of vascular networks that are lumenized, including close associations between pericytes and endothelial cells, the deposition of a basement membrane and interactions with astrocytes. Further developments of the iBRB-on-a-chip model could include incorporation of additional retinal cell types that interact directly or indirectly with the inner microvasculature, such as Müller cells, microglia, immune cell types and/or neuronal cell types.

We clarified this point page 17 lines 381-383 of the revised manuscript and have added the following sentence in the Discussion:

“The iBRB-on-a-chip incorporates retinal microvascular cell types that are most directly involved in NPDR pathogenesis, and additional cell types could be incorporated such as Müller glia, microglia, immune and neuronal cell types.”

3. If applicable, could the authors provide the references for the growing medium conditions used in the study? if the study employs a novel method for the growing medium condition, could you please provide details on the optimization process for each cell type and the tri-culture system used?

For regular cell culture, we used the media recommended by the suppliers. HRMVECs were cultured in EGM-2 MV Microvascular Endothelial Cell Growth Medium-2 BulletKit (EGM-2 MV, Lonza, CC-3202); HRP were cultured in Pericyte Medium (PM, ScienCell, 1201); and HRA were cultured in Astrocyte Medium (AM, ScienCell, 1801). In the chip, we used basal EGM-2 MV (Lonza) supplemented with the astrocyte growth supplements provided with the Astrocyte Medium (ScienCell). The tri-culture medium composition was based on the protocol for the blood-brain barrier-on-a-chip model adjusted to our basal media (*Campisi, M. et al 2018*).

4. Could the authors provide a justification for selecting TNF and IL-6 to mimic the diabetic condition in the study? It would be helpful if the authors could comment on the potential impact of excluding pro-inflammatory mediators on the observed effects.

We thank the reviewer for this question. Vitreous and plasma levels of TNF- α and IL-6 are elevated in patients with DR and these pro-inflammatory cytokines were added to the high glucose medium to mimic the inflammatory state observed in diabetes. A similar medium composition was investigated in an in vitro model of vascular organoids to induce features of diabetic vasculopathy (*Wimmer, R. A. et al 2019*). The authors used 75 mM glucose combined with 1ng/ml TNF- α and 1ng/ml IL-6 and observed clinically relevant phenotypes in the organoids such as basement membrane thickening. In our model, we applied the same concentrations of TNF- α and IL-6 as described in the article (1 ng/ml), and reduced the glucose concentration to 30 mM to be comparable to other in vitro studies on human retinal microvascular endothelial cells (*Rochfort, K. D. et al 2019; Eyre, J. J., et al 2020*) and to stay within physiological hyperglycemia concentrations.

We have clarified the choice of the diabetic cocktail composition in the manuscript (page 9 lines 173-176):

“Vitreous and plasma levels of TNF- α and IL-6 are elevated in patients with DR and these pro-inflammatory cytokines were added to the high glucose medium to mimic the inflammatory state observed in diabetes⁴⁸⁻⁵¹.”

Finally, to assess the impact of excluding TNF- α and IL-6 from the diabetic cocktail on the observed phenotypes, we have performed additional experiments where we treated the iBRB-on-a-chip with glucose, TNF- α or IL-6 and compared to the diabetic cocktail (new Extended Data Fig. 8). Treatment with glucose induced a decrease in vascular area and pericyte area comparable to the effects of the diabetic cocktail at D14. The fraction of acellular collagen was increased in the glucose condition compared to the diabetic group at the same time point (D14). At D28 however, the effects of the diabetic cocktail were much stronger than the glucose alone. In addition, diabetic treatment showed the highest detected levels of COL IV, IL-6 and ANG2, similar to TNF- α treatment. Interestingly, IL-6 treatment on D14 and all treatments on D28 showed increased levels of ANG1, while ANG1 levels remained unchanged in diabetic conditions. These data might suggest that high glucose might affect the vasculature in the early phases and that the combination with pro-inflammatory cytokines induces further cumulative damage.

We have included these observations in the manuscript (page 10 lines 202-207):

“First, evaluation of the effects of each individual component of the diabetic cocktail compared to combined effects showed significant decrease of vascular area and pericyte coverage, and increase of ghost vessels in diabetic conditions compared to glucose, IL-6, and TNF- α alone on D28 (Extended Data Fig. 8). These results demonstrate that the effects we observed in diabetic conditions are attributed to the combination of factors and not to the effect of one single component.”

Extended Data Fig. 8 | Effects of diabetic cocktail components. a, Quantification of vascular area, pericyte area and coverage, and fraction ghost vessels for untreated, diabetic, glucose, IL-6 and TNF- α conditions on D14 and D28. $n = 74$ untreated D7, $n =$

45 D14, n = 44 D28, n = 44 diabetic D14, n = 43 D28, n = 44 glucose D14, n = 44 D28, n = 45 IL-6 D14, n = 44 D28, n = 59 TNF- α D14 and n = 75 D28 treated networks analyzed from n = 3 replicate channels. **b**, Analyte measurements showing supernatant concentrations of human collagen IV alpha I, IL-6, Angiopoietin-1 and Angiopoietin-2 obtained from n = 3 replicate channels. Data are mean \pm s.d. *P<0.05; **P<0.01; ***P<0.001; ****P<0.0001; one-way ANOVA. **c**, Representative images of endothelial networks (UEA I) and overlay images with pericytes (PDGFR β) or basement membranes (COL IV) on D14 and D28, following 7 or 21 days of treatment respectively. Treatments with D-Glucose, IL-6 or TNF- α were tested individually, and in combination to produce the diabetic treatment (Diabetic). All images show maximum intensity projections of 245 μ m Z-stacks. Scale bars, 100 μ m (**c**).

5. Please provide data on cell viability data at days 7, 14, and 28 of the experiment. Additionally, did the diabetic cell culture condition induce more cell death when compared to the control condition?

We thank the reviewer for this comment and have performed cell viability experiments at D9, D11, D14, and D28 (new Extended Data Fig. 9). CellEvent staining showed that the viability remained above 95% in untreated and IL-6-treated conditions, and above 85% in diabetic conditions. Toxicity was thus limited, but increased significantly in the presence of diabetic cocktail at all time points, which is in agreement with the observed phenotypes. LDH release as a measure of cell membrane integrity was also increased in diabetic conditions compared to the untreated control and IL-6 treatment, albeit not significantly.

We have added the following paragraph describing the results in the manuscript (page 10 lines 207-209):

“Then, an apoptosis (Caspase-3/7) detection assay showed that viability remained above 95% in untreated and IL-6-treated conditions, and above 85% in diabetic conditions, indicating increased cell death (Extended Data Fig. 9).”

A method section was added for cell viability assays on page 24 and 25 lines 551-564.

Extended Data Fig. 9 | Cell viability assays. a, Quantification of the percentage of apoptotic cells (CellEvent) normalized to total nuclei (DAPI) per field for untreated, diabetic and IL-6-treated conditions over time. $n = 45$ untreated D9, D11, D14 and D28, $n = 45$ diabetic D9 and D14, $n = 44$ diabetic D11 and D28, $n = 45$ IL-6 D9, D14 and D28, and $n = 44$ IL-6 D11 treated networks analyzed from $n = 3$ replicate channels. **b**, Quantification of LDH release for untreated, diabetic and IL-6 conditions over time obtained from $n = 3$ replicate channels for each condition. Data are mean \pm s.d. * $P < 0.05$; ** $P < 0.01$; *** $P < 0.001$; **** $P < 0.0001$; one-way ANOVA. **c**, Representative images of untreated control and Staurosporine-treated cells on D7. CellEvent marks apoptotic nuclei and DAPI all nuclei. Staurosporine is a positive control inducing apoptosis. **d**, Representative images of untreated control, diabetic

cocktail, IL-6 alone, and Staurosporine control on D9, D11, D14 and D28. All images show maximum intensity projections of 245 μm Z-stacks. Scale bars, 100 μm (c, d).

6. Please provide additional information or references to justify the selection of DHDP and AGE to study pericyte-endothelial cell interactions?

19,20-dihydroxydocosapentaenoic acid (or DHDP) is elevated in retina and vitreous humor of patients with DR. It has been described that DHDP can perturb endothelial cell-pericyte interactions and endothelial cell junctions by integrating in the cell membrane, inhibiting the catalytic unit of the gamma-secretase and disrupting the localization of the Cadherins (Hu, J. et al. Inhibition of soluble epoxide hydrolase prevents diabetic retinopathy. *Nature* 552, 248–252 (2017)). After 7 days of treatment with DHDP, we observed reduction of the vascular area, pericyte area and pericyte coverage compared to the untreated group (Fig. 5c-e). These observations confirmed the interactions of DHDP with endothelial cell junctions and endothelial cell-pericyte interactions.

Advanced Glycation End products (AGE) are generated by the non-enzymatic glycation of proteins, lipids or amino acids by high glucose and accumulation of AGE is a characteristic of diabetic retinopathy. Binding of AGE to its receptor can lead to pericyte apoptosis via activation of the NF-kB signaling (Kang, C. et al. Tolerogenic dendritic cells and TLR4 / IRAK4 / NF- k B signaling pathway in allergic rhinitis. 1–10 (2023)).

We have clarified the rationale about the selection of DHDP and AGE in the manuscript and have added the references (page 12 and 13 lines 267-272).

“We also applied 19,20-dihydroxydocosapentaenoic acid (DHDP) and advanced glycated end product (AGE) treatments expected to interfere with homeostatic cellular processes (Fig. 5a). DHDP can perturb endothelial cell-pericyte interactions and endothelial cell junctions by integrating in the cell membrane, inhibiting the catalytic unit of gamma-secretase and disrupting the localization of cadherins⁵⁸. AGE accumulation in DR can lead to pericyte apoptosis via activation of the NF-kB signaling⁵⁹.”

7. Would it be possible to utilize TUNEL or Annexin V assays to identify the initial cell type undergoing apoptosis in the diabetic condition of the tri-culture system? It would be interesting to know if this approach could provide insight into the sequence of events in non-proliferative DR and facilitate the discovery of potential therapeutic targets

We agree with the reviewer that understanding the sequence of events in non-proliferative DR would provide additional hypotheses on disease mechanisms and may facilitate new target identification and we are also very interested in this aspect.

We performed an apoptosis assay (CellEvent) at D9, D11 and D14, and co-stained with PDGFRb and UEA I (new Extended Data Fig. 10). We observed apoptotic signals overlapping both with the vessels and with pericytes. We counted and plotted apoptotic cells overlapping

with either PDGFR β (pericytes PC) or UEA I (endothelial cells EC). Based on these results, endothelial cells are the initial cell type undergoing apoptosis on D9, followed by pericytes from D11. The number of apoptotic ECs increased in all conditions, and was more than 2-fold greater in diabetic conditions. This suggests that ECs are either more vulnerable to the diabetic cocktail than pericytes and become apoptotic first, or EC loss triggers pericyte apoptosis.

We have added the following paragraph describing the results in the manuscript (page 10 lines 209-215):

“In order to provide insight into the sequence of events during initiation of NPDR, we co-stained apoptotic nuclei with endothelial UEA I and pericyte PDGFR β markers. Apoptotic cells on D9, D11 and D14 during early treatment overlapped mostly with UEA I+ vascular networks, suggesting that endothelial cells are the initial cell type undergoing apoptosis on D9, followed by pericytes from D11 (Extended Data Fig. 10a). The number of apoptotic ECs increased in all conditions, and was more than 2-fold greater in diabetic conditions (Extended Data Fig. 10b).”

Extended Data Fig. 10 | Sequence of cell death events. a, Representative images of apoptotic nuclei (CellEvent), endothelial network (UEA I) and pericyte (PDGFR β) stainings on D9, D11 and D14 for untreated control, diabetic, IL-6, and Staurosporine-treated conditions (positive control). b, Number of apoptotic cells by cell type; endothelial cells (EC) in orange and pericytes (PC) in green on D9, D11 and D14 for untreated control, diabetic and IL-6 conditions. One representative ROI per n = 3 replicate channels was quantified for each condition on D9, D11 and D14. All images show maximum intensity projections of 245 μ m Z-stacks. Scale bars, 100 μ m (a).

8. Could you provide information on previous studies that have utilized on-chip in vitro models with retinal cells to mimic the blood-retinal barriers? Mentioning these studies would help to emphasize the innovative approach of the current study.

We thank the reviewer for this comment and we have added a paragraph about the current in vitro models of the iBRB and outer BRB in the Introduction (page 5 lines 89-100).

“Several organ-on-a-chip technologies have been leveraged for ophthalmology applications to model the choroid, the outer BRB, the interface between photoreceptors and RPE, or investigate pericytes-endothelial cell interactions^{29–33}. There is currently no human model system to study pathophysiological changes in cell types of the iBRB. The models of the iBRB that have been developed so far are based on flat Transwell inserts where endothelial cells, pericytes and/or astrocytes are seeded on a polycarbonate membrane or at the bottom of the well^{34–36}. While these Transwell iBRB models have highlighted the contribution of astrocytes to endothelial barrier properties, they lack an extracellular matrix, direct physical contact between the cells and a vascular-like architecture. Reconstructing these specialized features of the iBRB in vitro and interrogating them experimentally will enable better understanding of iBRB properties in physiological and disease conditions as well as provide a platform for validating therapeutic approaches.”

Reviewer #4 (Remarks to the Author):

Maurissen et al report an interesting and novel creation of an in vitro model for the blood-retinal barrier (BRB). The authors combine human retinal microvascular cells, pericyte, and astrocytes in a fibrin gel with microfluidics to extremely well characterize a microvascular endothelial network that includes lumens, tight junctions, pericyte coverage, and low permeability. Next, the authors create a diabetes model, including high glucose, IL6, and TNFa, and show that diabetes to capillary dropout, pericyte loss, and ghost vessels in a thorough manner. These alterations are accompanied by gene and protein express changes that mimic what is known about gene expression alterations in early NPDR. Finally, the authors disrupt EC-pericyte interactions through multiple mechanisms and recapitulate aspects of early NPDR reported in vivo. The manuscript is novel, well-written, and very thorough. My major concern is the very high non-physiologic IL6/TNFa concentrations used to create the diabetes model.

Major Comments:

1. IL6 levels in aqueous humor and vitreous humor have been previously reported. In the aqueous humor (PMID: 22511846), IL6 levels range from 1.9-16.4 pg/ml in control and DR eyes while TNFa levels were 0-7.6 pg/ml. In the vitreous humor (PMID: 19997642), IL6 levels range from 12-330 pg/ml in control and PDR eyes, while TNFa levels were not detectable. The authors use 1 ng/ml IL6 and TNFa in their diabetes model, which are greater than physiologic or pathophysiologic levels. In extended Fig 10, it is clear that IL6 alone can reduce pericyte coverage, where TNFa alone has little effect. Based upon this knowledge and extended Fig 10, I am concerned that the “diabetes model” might be more of a model of supraphysiologic IL6-induced pericyte toxicity, which would significantly hamper the conclusions of this study.

We thank the reviewer for thorough feedback on physiological conditions.

In patients, NPDR manifests over several years and lower doses of cytokines are exerting effects over a period of time that is longer to what we can model in vitro (here we treated for several weeks). Therefore we cannot expect a direct correlation between in vivo concentration and in vitro dosing. We selected a glucose concentration that is in the upper range of physiological hyperglycemia grade 4. The concentrations of IL-6 and TNF- α were based on a publication that used these cytokines to model diabetic vasculopathy on vascular organoids (Wimmer, R. A. et al 2019).

We do understand the concern of the reviewer about modeling supraphysiologic IL-6 effects and we have compared cell viability between untreated, diabetic and IL-6 groups (new Extended Data Fig. 9). While we observed an increase in apoptotic cell numbers (measured by CellEvent) in the diabetic group compared to IL-6 and untreated groups, the viability of IL-6 and untreated groups were similar. We also performed a LDH to quantify cell membrane integrity and while there was an increase of LDH in the diabetic group compared to untreated and IL-6 groups, the difference was not significant. These results show that IL-6 alone at 1 ng/ml is not mediating any toxicity on the iBRB-on-a-chip model and that the effect observed in the diabetic group are due to the combination of the different factors.

We have added the following paragraph in the manuscript (page 10 lines 207-209):

“Then, an apoptosis (Caspase-3/7) detection assay showed that viability remained above 95% in untreated and IL-6-treated conditions, and above 85% in diabetic conditions, indicating increased cell death (Extended Data Fig. 9).”

Extended Data Fig. 9 | Cell viability assays. *a*, Quantification of the percentage of apoptotic cells (CellEvent) normalized to total nuclei (DAPI) per field for untreated, diabetic and IL-6-treated conditions over time. $n = 45$ untreated D9, D11, D14 and D28, $n = 45$ diabetic D9 and D14, $n = 44$ diabetic D11 and D28, $n = 45$ IL-6 D9, D14 and D28, and $n = 44$ IL-6 D11 treated networks analyzed from $n = 3$ replicate channels. *b*, Quantification of LDH release for untreated, diabetic and IL-6 conditions over time obtained from $n = 3$ replicate channels for each condition. Data are mean \pm s.d. * $P < 0.05$; ** $P < 0.01$; *** $P < 0.001$; **** $P < 0.0001$; one-way ANOVA. *c*, Representative images of untreated control and Staurosporine-treated cells on D7. CellEvent marks apoptotic nuclei and DAPI all nuclei. Staurosporine is a positive control inducing apoptosis. *d*, Representative images of untreated control, diabetic cocktail, IL-6 alone, and

Staurosporine control on D9, D11, D14 and D28. All images show maximum intensity projections of 245 μm Z-stacks. Scale bars, 100 μm (c, d).

In addition, we have evaluated the effects of each individual component of the diabetic cocktail compared to combined effects (new Extended Data Fig. 8). We observed a significant decrease of vascular area and pericyte coverage in the DR group compared to glucose, IL-6, and TNF- α groups at D28. The fraction of acellular collagen increased in the DR group compared to the other groups. These results demonstrate that the effects we observed with the DR cocktail are attributed to the combination of factors and not to the effect of one single component.

We have added the following paragraph describing the results (page 10 lines 201-207).

“Additionally, we investigated further effects of the diabetic treatment between D7 and D28. First, evaluation of the effects of each individual component of the diabetic cocktail compared to combined effects showed significant decrease of vascular area and pericyte coverage, and increase of ghost vessels in diabetic conditions compared to glucose, IL-6, and TNF- α alone on D28 (Extended Data Fig. 8). These results demonstrate that the effects we observed in diabetic conditions are attributed to the combination of factors and not to the effect of one single component.”

Extended Data Fig. 8 | Effects of diabetic cocktail components. a, Quantification of vascular area, pericyte area and coverage, and fraction ghost vessels for untreated, diabetic, glucose, IL-6 and TNF- α conditions on D14 and D28. $n = 74$ untreated D7, $n =$

45 D14, n = 44 D28, n = 44 diabetic D14, n = 43 D28, n = 44 glucose D14, n = 44 D28, n = 45 IL-6 D14, n = 44 D28, n = 59 TNF- α D14 and n = 75 D28 treated networks analyzed from n = 3 replicate channels. **b**, Analyte measurements showing supernatant concentrations of human collagen IV alpha I, IL-6, Angiopoietin-1 and Angiopoietin-2 obtained from n = 3 replicate channels. Data are mean \pm s.d. *P<0.05; **P<0.01; ***P<0.001; ****P<0.0001; one-way ANOVA. **c**, Representative images of endothelial networks (UEA I) and overlay images with pericytes (PDGFR β) or basement membranes (COL IV) on D14 and D28, following 7 or 21 days of treatment respectively. Treatments with D-Glucose, IL-6 or TNF- α were tested individually, and in combination to produce the diabetic treatment (Diabetic). All images show maximum intensity projections of 245 μ m Z-stacks. Scale bars, 100 μ m (c).

2. In figure 2e, the authors measure permeability of their microvascular network using TRITC-labeled 70 kDa dextran, which is very cool. The authors show that TNF α increases permeability in the binary mask. From the original image, it is difficult for me to appreciate the leakage shown in the binary mask. Can this image quality be improved so the leakage is apparent? Can we show a control for comparison? Why wasn't this important assay performed in the "diabetes" model?

We thank the reviewer for this comment. We have improved the image of the TNF- α treated condition so that it is easier to observe the leakage of the fluorescent dextran. In addition, we have added images on the untreated control (Fig. 2e).

Fig. 2 e, Representative image of control and TNF- α -treated (1 ng ml⁻¹, 24 h) iBRB MVNs perfused on D10 with TRITC-labelled dextran (100 μ g ml⁻¹, 70 kDa), acquired by fluorescence confocal microscopy at 5 min intervals. At t = 10 min, the binary mask shows leakage for the TNF- α -treated condition. Perfusion images are maximum intensity projections of 290 μ m Z-stacks. **f**, Apparent permeability coefficients were quantified on D10 in untreated and TNF- α -treated MVNs using 70 kDa TRITC-dextran as a tracer. n = 6 untreated and n = 3 TNF- α -treated networks. Data are mean \pm s.d. **P = 0.0029; two-tailed Student's t-test. Scale bars, 100 μ m.

We also calculated the permeability to 70 kDa dextran in diabetic and control conditions at D14 and D28 using three different batches of astrocytes in the tri-culture, which greatly influenced long-term perfusion efficacy. MVNs formed with the Human Retinal Astrocytes (HRA) batch#1 were not perfusable in control and diabetic conditions D14 and D28, thus we could not calculate the permeability coefficients (new Extended Data Fig. 11a top row). MVNs formed with the HRA batch#2 were perfusable in the control group while perfusion was strongly reduced in the diabetic group at D14. The permeability coefficient of the remaining perfusable diabetic vessels were comparable to the untreated control (new Extended Data Fig. 11d). Perfusability of MVNs formed with HRA batch#2 was insufficient in control and diabetic conditions on D28 to be able to quantify permeability coefficients. Finally, MVNs formed with Human Cortical Astrocytes were perfusable until D28 in control conditions while reduced perfusion was observed in the diabetic group. The permeability was slightly increased in the diabetic group compared to the control group at D28.

These observations are in line with the literature where increase in VEGF and permeability are a consequence of vascular regression.

We have added the following sentences in the Manuscript (page 10 and 11 lines 215-224):

“Finally, we perfused untreated and diabetic MVNs with fluorescent-labelled dextran (70kDa TRITC-dextran) on D14 and D28. While the current tri-culture setup was not perfusable at these later time points, we identified other astrocyte batches that supported long-term perfusion and produced similar diabetic phenotypes on D28 (Extended Data Fig. 11a, b). Diabetic treatment strongly reduced perfusability, the percentage of TRITC+ perfusable vessels compared to UEA I+ total vessels for all astrocyte batches (Extended Data Fig. 11c). In perfusable networks, we quantified vascular permeability and did not observe increased leakage in perfusable diabetic vessels (Extended Data Fig. 11d). In sum, diabetic treatment is responsible for disease phenotypes, and causes increased cell death and decreased perfusability that contribute to iBRB dysfunction.”

The permeability assay method section was completed on page 25 lines 574-580.

Extended Data Fig. 11 | Perfusability and permeability assays. *a*, Images of untreated control and diabetic iBRB MVNs perfused with TRITC-labelled 70kDa dextran at $t = 0$ and $t = 10$ min. Human retinal astrocytes (HRA) and human cortical astrocytes (HCA) from different donors were compared on D14 and D28. Triplicate channels are shown for each condition. Images show maximum intensity projections of $190 \mu\text{m}$ Z-stacks. Channel height visible is 1.3 mm . *b*, Quantifications of vascular area, pericyte area and coverage, and COL IV area and avascular area. $n = 45$ HRA#1 untreated, $n = 45$ diabetic D28, $n = 45$ HRA#2 untreated, $n = 45$ diabetic D28, $n = 91$ HCA untreated, $n = 90$ diabetic D28 treated networks analyzed from $n = 3$ replicate channels. $n = 91$ HCA diabetic D28 treated networks for pericyte area. $n = 30$ HRA#1 untreated, $n = 30$ diabetic D28, $n = 30$ HRA#2 untreated, $n = 30$ diabetic D28, $n = 60$ HCA untreated, $n = 58$ diabetic D28

treated networks for COL IV area and avascular area. Data are mean \pm s.d. * $P < 0.05$; ** $P < 0.01$; *** $P < 0.001$; **** $P < 0.0001$; one-way ANOVA. **c**, Perfusability corresponding to the fraction of TRITC+ perfused vessels to UEA I+ total vessels on D28 for $n = 3$ HRA#2 and $n = 6$ HCA replicate channels per condition. **d**, Permeability values of perfusable untreated and diabetic MVNs containing HRA#2 or HCA in tri-cultures. $n = 9$ HRA#2 (3 channels), and $n = 18$ HCA (6 channels) network ROI analyzed per condition.

3. The authors state that, “Addition of astrocytes reduced the vascular area and diameter, making the networks more microvascular-like” – I do not see any data on diameter, can this be quantified like vascular area?

We agree with the reviewer that the diameter is an interesting parameter to characterize vascular morphology and we have included measurements of the vascular diameter to the quantification panel (Extended Data Fig. 1f, g; page 7 line 133).

On the left figure, the mean diameter per channel was quantified by dividing the total vascular area by the total skeleton length on 3 replicate channels from independent experiments. The figure on the right displays the distribution of vessel diameter within one channel quantified manually. Both image analyses show the vessel diameter was reduced when astrocytes and pericytes were added to the endothelial cells.

Minor Concerns:

1. The authors state: “HRMVECs isolated from different donors did not all lead to the formation of MVNs” in the results, and “Original vials of primary Human Retinal Microvascular Endothelial Cells (HRMVEC, Cell Systems, ACBRI 181) were obtained at passage 3 (P3), Human Retinal Pericytes (HRP, Cell Systems, ACBRI 183) at P3 and Human Retinal Astrocytes (HRA, ScienCell, 1870) at P1” in the methods. Does this mean that the authors purchased cells

from different donors and compared them? I don't see any mention of this possibility on their website and want to make sure that I, and the future reader, understand correctly. Is this merely cell quality?

We acknowledge that information about the different cell types used in the study was not clearly detailed in the original version of the manuscript. To clarify this to the reader, we have included in the Supplementary Information a Table summarizing for each cell type used in this study, its source, type, supplier, catalog number, batch number as well as information about the donor, when made available by the company. This list represents all HRMVEC, HRP, and HRA batches and donors that were commercially available at the time of this study.

Extended Data Table I: Information about cells used in this study.

Cell type	Source	Type	Supplier	Cat number	Batch number	Donor information
HRMVEC - 1	retina	primary	Cell Sytems	ACBRI 181	181.04.01.02.02	NA
HRMVEC - 2	retina	primary	Pelo-Biotech	PB-CH-160-8511	QC-18B19F09	28 year-old female
HRMVEC - 3	retina	primary	Pelo-Biotech	PB-CH-160-8511	QC-03B18F04	NA
HRMVEC - 4	retina	primary	Cell Biologics	H-6065	F021518Ag	NA
HRMVEC - 5	retina	primary	Cell Biologics	H-6065	M021518	NA
HRMVEC - 6	retina	primary	Cell Biologics	H-6065	122118U	NA
HRMVEC - 7	retina	primary	Cell Biologics	H-6065	120117Ag	NA
HRP	retina	primary	Cell Sytems	ACBRI 183	183.02.01.01.10	NA
HRA - 1	retina	primary	ScienCell	1870	23521	20 week-old male
HRA - 2	retina	primary	ScienCell	1870	30020	56 year-old female
HCA	brain	immortalized	Innoprot	P10251-IM	NA	NA

NA: not provided by the supplier

References

- Al Ahmad, A., Taboada, C. B., Gassmann, M. & Ogunshola, O. O. Astrocytes and pericytes differentially modulate blood-brain barrier characteristics during development and hypoxic insult. *J. Cereb. Blood Flow Metab.* 31, 693–705 (2011)
- Atli, H., Onalan E., Yakar B., D. Duzenci D., & Dönde, E. Predictive value of inflammatory and hematological data in diabetic and non-diabetic retinopathy. *Eur. Rev. Med. Pharmacol. Sci.* 26, 76–83 (2022)
- Bonkowski, D., Katyshev, V., Balabanov, R. D., Borisov, A. & Dore-Duffy, P. The CNS microvascular pericyte: Pericyte-astrocyte crosstalk in the regulation of tissue survival. *Fluids Barriers CNS* 8, 1–12 (2011)
- Bryan, B. A. & D'Amore, P. A. Pericyte isolation and use in endothelial/pericyte coculture models. *Methods Enzymol.* 443, 315–331 (2008)
- Campisi, M. et al. 3D self-organized microvascular model of the human blood-brain barrier with endothelial cells, pericytes and astrocytes. *Biomaterials* 180, 117–129 (2018)
- Daruich, A. et al. Mechanisms of macular edema: Beyond the surface. *Prog. Retin. Eye Res.* 63, 20–68 (2018)
- Demircan, N., Safran, B. G., Soylu, M., Ozcan, A. A. & Sizmaz, S. Determination of vitreous interleukin-1 (IL-1) and tumour necrosis factor (TNF) levels in proliferative diabetic retinopathy. *Eye* 20, 1366–1369 (2006)
- Díaz-Flores, L. et al. Pericytes. Morphofunction, interactions and pathology in a quiescent and activated mesenchymal cell niche. *Histol. Histopathol.* 24, 909–969 (2009)
- Eyre, J. J., Williams, R. L. & Levis, H. J. A human retinal microvascular endothelial-pericyte co-culture model to study diabetic retinopathy in vitro. *Exp. Eye Res.* 201, 108293 (2020)
- Fresta, C. G. et al. A new human blood–retinal barrier model based on endothelial cells, pericytes, and astrocytes. *Int. J. Mol. Sci.* 21, 1–17 (2020)
- Klaassen, I., Van Noorden, C. J. F. & Schlingemann, R. O. Molecular basis of the inner blood-retinal barrier and its breakdown in diabetic macular edema and other pathological conditions. *Prog. Retin. Eye Res.* 34, 19–48 (2013)
- Rochfort, K. D. et al. COMP-ang1 stabilizes hyperglycemic disruption of blood-retinal barrier phenotype in human retinal microvascular endothelial cells. *Investig. Ophthalmol. Vis. Sci.* 60, 3547–3555 (2019)
- Sánchez-Palencia, D. M., Bigger-Allen, A., Saint-Geniez, M., Arboleda-Velásquez, J. F. & D'Amore, P. A. Coculture Assays for Endothelial Cells-Mural Cells Interactions. *Tumor Angiogenesis Assays: Methods and Protocols* 35–47 (2016)

Wimmer, R. A., Leopoldi A., Aichinger M., Wick N., Hantusch B., Novatchkova M., Taubenschmid J., Hämmerle M., Esk C., Bagley J.A., Lindenhofer D., Chen G., Boehm M., Agu C.A., Yang F., Fu B., Zuber J., Knoblich J.A., Kerjaschki D. & Penninger J.M., Human blood vessel organoids as a model of diabetic vasculopathy. Nature 565, 505–510 (2019)

Wisniewska-Kruk, J. et al. A novel co-culture model of the blood-retinal barrier based on primary retinal endothelial cells, pericytes and astrocytes. Exp. Eye Res. 96, 181–190 (2012)

Yao, H. et al. The development of blood-retinal barrier during the interaction of astrocytes with vascular wall cells. Neural Regen. Res. 9, 1047–1054 (2014)

REVIEWER COMMENTS

Reviewer #1 (Remarks to the Author):

The authors carefully tried to respond to this reviewers' comment. However, the authors could not adequately answer the critical question why VEGF and angiopoietin 2 were not applied to this authors' chip. VEGF and angiopoietin 2 have been known to be major growth factors in diabetic retinopathy. To properly validate and convince the usefulness of the authors' chip, the data should be provided.

Reviewer #2 (Remarks to the Author):

The authors have addressed all my comments, and I have no further comments.

Reviewer #3 (Remarks to the Author):

The reviewer is satisfied with the authors' answers and commends their hard work in addressing the comments. The reviewer has no further questions.

Reviewer #4 (Remarks to the Author):

The authors have satisfied my concerns and improved the quality of the manuscript. Congratulations on a very interesting body of work.

REVIEWER COMMENTS

Reviewer #1 (Remarks to the Author):

The authors carefully tried to respond to this reviewers' comment. However, the authors could not adequately answer the critical question why VEGF and angiopoietin 2 were not applied to this authors' chip. VEGF and angiopoietin 2 have been known to be major growth factors in diabetic retinopathy. To properly validate and convince the usefulness of the authors' chip, the data should be provided.

Thank you for taking the time to re-review our manuscript. While at Roche we are keenly interested in VEGF/Ang-2 biology in the context of diabetic retinopathy, we respectfully disagree that these factors should be included in the diabetic induction cocktail in this model for the following reason: we model the early events in diabetic retinopathy prior to the proliferative stage, when Ang-2 and VEGF reach peak levels.¹⁻⁴

According to data from biomarker studies published by independent labs, VEGF and Ang-2 are upregulated in diabetic retinopathy ocular fluids, but do not reach peak levels until more advanced stages of disease (likely in response to retinal ischemia).¹⁻⁴ The upregulation of VEGF/Ang-2 activates hallmark of pro-angiogenic processes characteristic of proliferative diabetic retinopathy (PDR) that result in excessive vascular permeability and neovascularization.^{5,6} In addition to VEGF and Ang-2, many other growth factors are also upregulated in PDR and likely contribute to the phenotype.^{4,7}

Regarding the staging of the disease model, the present study was designed to mimic early pathophysiological phenotypes of diabetic retinopathy, i.e., non-proliferative diabetic retinopathy (NPDR) characterized by pericyte loss and vasoregression, that precede ischemia and neovascularization or PDR, which is characterized by angiogenesis and proliferation of new vessels. Therefore, to model early events we supplemented the model with IL-6 and TNF- α in addition to high glucose as these factors are known to mediate inflammation in the early phases of DR, prior to angiogenesis and PDR.^{8,9} Pathophysiological changes observed in our model using these treatment conditions resembled clinical hallmarks of early DR.

While we have demonstrated in this study the usefulness of the iBRB-on-a-chip model to recapitulate early phenotypes of DR, this model has broader implications. Beyond what is described in this manuscript, the platform could be used by others groups to study later phases of the diseases, such as PDR for example, to test additional combinations of disease triggers and/or to assess potential therapies at different disease stages.

¹Adamis A.P., Miller J.W., Brelan M.T., et al. (1994) Increased vascular endothelial growth factor levels in the vitreous of eyes with proliferative diabetic retinopathy. *American Journal of Ophthalmology* 118(4):445-50. doi: 10.1016/s0002-9394(14)75794-0

²Aiello L.P., Avery R.L., Arrigg P.G., et al. (1994) Vascular Endothelial Growth Factor in Ocular Fluid of Patients with Diabetic Retinopathy and Other Retinal Disorders. *The New England Journal of Medicine* 1;331(22):1480-7. doi: 10.1056/NEJM199412013312203

³Watanabe D., Suzuma K., Suzuma I., et al. (2005) Vitreous levels of angiopoietin 2 and vascular endothelial growth factor in patients with proliferative diabetic retinopathy. *American Journal of Ophthalmology* 139(3):476-81. doi: 10.1016/j.ajo.2004.10.004

⁴Tsai T., Alwees M., Asaad M.A., et al. (2023) Increased Angiopoietin-1 and -2 levels in human vitreous are associated with proliferative diabetic retinopathy. *PLoS ONE* 18(1): e0280488. <https://doi.org/10.1371/journal.pone.0280488>

⁵Cheung N., Mitchell P., Wong T.Y., (2010) Diabetic retinopathy. *The Lancet* 376: 124–36. [https://doi.org/10.1016/S0140-6736\(09\)62124-3](https://doi.org/10.1016/S0140-6736(09)62124-3)

⁶Wong, T., Cheung, C., Larsen, M. et al. (2016) Diabetic retinopathy. *Nature Reviews Disease Primers* 2, 16012. <https://doi.org/10.1038/nrdp.2016.12>

⁷Mason, R.H., Minaker, S.A., Lahaie Luna, G. et al. (2022) Changes in aqueous and vitreous inflammatory cytokine levels in proliferative diabetic retinopathy: a systematic review and meta-analysis. *Eye*. <https://doi.org/10.1038/s41433-022-02127-x>

⁸Rübsam A., Parikh S., Fort P.E. (2018) Role of Inflammation in Diabetic Retinopathy. *International Journal of Molecular Sciences* 22;19(4):942. doi: 10.3390/ijms19040942

⁹Semararo F., Cancarini A., dell’Omo R. et al. (2015) Diabetic retinopathy: vascular and inflammatory disease. *Journal of Diabetes Research* 2015:582060 <https://doi.org/10.1155/2015/582060>

REVIEWERS' COMMENTS

Reviewer #3 (Remarks to the Author):

I agree with Reviewer 1's comment that adding VEGF and Ang-2 to the chip model would enhance its usefulness and clinical applicability. However, as the study primarily focuses on modelling the early stages of diabetic retinopathy, which mainly involves the proinflammatory phase, including proangiogenic factors may be beyond the scope of the study.

Nevertheless, exploring the effects of VEGF and Ang-2 on the chip model in future studies would be a valuable avenue of research. Future studies could also assess the potential for modelling neovascularization formation on the chip, which would greatly contribute to our understanding of disease progression during the proliferative phase.

Despite its limitations, the current study represents a significant advancement in creating a more clinically relevant model using human cells to study diabetic retinopathy. These models hold great potential in discovering new treatments and validating the translational applicability of potential new drugs.